# Non-parametric Outlier Synthesis

**Leitian Tao**
School of Information Engineering
Wuhan University
taoleitian@gmail.com

**Xuefeng Du, Xiaojin Zhu, Yixuan Li**
Department of Computer Sciences
University of Wisconsin - Madison
{xfdu,jerryzhu,sharonli}@cs.wisc.edu

## Abstract

Out-of-distribution (OOD) detection is indispensable for safely deploying machine learning models in the wild. One of the key challenges is that models lack supervision signals from unknown data, and as a result, can produce overconfident predictions on OOD data. Recent work on outlier synthesis modeled the feature space as parametric Gaussian distribution, a strong and restrictive assumption that might not hold in reality. In this paper, we propose a novel framework, *non-parametric outlier synthesis* (NPOS), which generates artificial OOD training data and facilitates learning a reliable decision boundary between ID and OOD data. Importantly, our proposed synthesis approach does not make any distributional assumption on the ID embeddings, thereby offering strong flexibility and generality. We show that our synthesis approach can be mathematically interpreted as a rejection sampling framework. Extensive experiments show that NPOS can achieve superior OOD detection performance, outperforming the competitive rivals by a significant margin. Code is publicly available at https://github.com/deeplearning-wisc/npos.

## 1 Introduction

When deploying machine learning models in the open and non-stationary world, their reliability is often challenged by the presence of out-of-distribution (OOD) samples. As the trained models have not been exposed to the unknown distribution during training, identifying OOD inputs has become a vital and challenging problem in machine learning. There is an increasing awareness in the research community that the source-trained models should not only perform well on the In-Distribution (ID) samples, but also be capable of distinguishing the ID vs. OOD samples.

To achieve this goal, a promising learning framework is to jointly optimize for both (1) accurate classification of samples from $\mathbb{P}_{in}$, and (2) reliable detection of data from outside $\mathbb{P}_{in}$. This framework thus integrates distributional uncertainty as a first-class construct in the learning process. In particular, an uncertainty loss term aims to perform a level-set estimation that separates ID vs. OOD data, in addition to performing ID classification. Despite the promise, a key challenge is *how to provide OOD data for training without explicit knowledge about unknowns*. A recent work by Du et al. (2022c) proposed synthesizing virtual outliers from the low-likelihood region in the feature space of ID data, and showed strong efficacy for discriminating the boundaries between known and unknown data. However, they modeled the feature space as class-conditional Gaussian distribution — a strong and restrictive assumption that might not always hold in practice when facing complex distributions in the open world. Our work mitigates the limitations.

In this paper, we propose a novel learning framework, **N**on-**P**arametric **O**utlier **S**ynthesis (**NPOS**), that enables the models learning the unknowns. Importantly, our proposed synthesis approach does not make any distributional assumption on the ID embeddings, thereby offering strong flexibility and generality especially when the embedding does not conform to a parametric distribution. Our framework is illustrated in Figure 1. To synthesize outliers, our key idea is to "spray" around the low-likelihood ID embeddings, which lie on the boundary between ID and OOD data. These boundary points are identified by non-parametric density estimation with the nearest neighbor distance. Then, the artificial outliers are sampled from the Gaussian kernel centered at the embedding of the boundary ID samples. Rejection sampling is done by only keeping synthesized outliers with low likelihood. Leveraging the synthesized outliers, our uncertainty loss effectively performs the level-

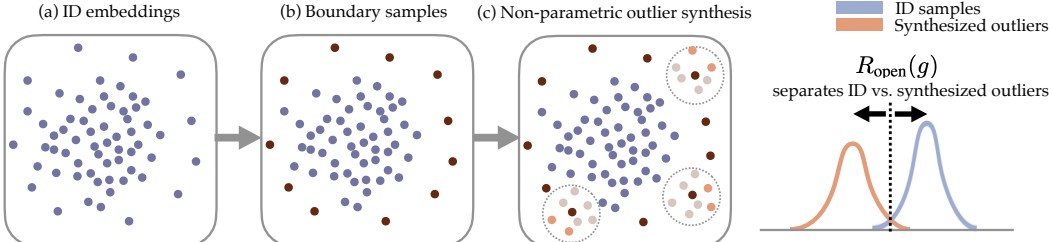

Figure 1: Illustration of our non-parametric outlier synthesis (**NPOS**). (a) Embeddings of ID data are optimized using Equation 8, which facilitates learning distinguishable representations. (b) Boundary ID embeddings are selected based on the non-parametric $k$-NN distance. (c) Outliers are synthesized by sampling from a multivariate Gaussian distribution centered around the boundary embeddings. Rejection sampling is performed by keeping the synthesized outliers (orange) with the lowest likelihood. The risk term $R_{\text{open}}$ performs level-set estimation, learning to separate the synthesized outliers and ID embeddings (Equation 6). Best view in color.

set estimation, learning to separate data between two sets (ID vs. outliers). This loss term is crucial to learn a compact decision boundary, and preventing overconfident predictions for unknown data.

Our uncertainty loss is trained in tandem with another loss that optimizes the ID classification and embedding quality. In particular, the loss encourages learning highly distinguishable ID representations where ID samples are close to the class centroids. Such compact representations are desirable and beneficial for non-parametric outlier synthesis, where the density estimation can depend on the quality of learned features. Our learning framework is end-to-end trainable and converges when both ID classification and ID/outlier separation perform satisfactorily.

Extensive experiments show that NPOS achieves superior OOD detection performance, outperforming the competitive rivals by a large margin. In particular, Fort et al. (2021) recently exploited large pre-trained models for OOD detection. They fine-tuned the model using standard cross-entropy loss and then applied the maximum softmax probability (MSP) score in testing. Under the same pre-trained model weights (*i.e.*, CLIP-B/16 (Radford et al., 2021)) and the same number of fine-tuning configuration, NPOS reduces the FPR95 from 41.87% (Fort et al., 2021) to 5.76% — a direct **36.11%** improvement. Since both methods employ MSP in testing, the performance gap signifies the efficacy of our training loss using outlier synthesis for model regularization. Moreover, we contrast NPOS with the most relevant baseline VOS (Du et al., 2022c) using parametric outlier synthesis, where NPOS outperforms by 13.40% in FPR95. The comparison directly confirms the advantage of our proposed non-parametric outlier synthesis approach.

To summarize our key contributions:

1. We propose a new learning framework, *non-parametric outlier synthesis* (NPOS), which automatically generates outlier data for effective model regularization and improving test-time OOD detection. Compared to a recent method VOS that relies on parametric distribution assumption (Du et al., 2022b), NPOS offers both stronger performance and generality.

2. We mathematically formulate our outlier synthesis approach as a rejection sampling procedure. By non-parametric density estimation, our training process approximates the level-set distinguishing ID and OOD data.

3. We conduct comprehensive ablations to understand the efficacy of NPOS, and further verify its scalability to a large dataset, including ImageNet. These results provide insights on the non-parametric approach for OOD detection, shedding light on future research.

## 2 PRELIMINARIES

**Background and notations.**    Over the last few decades, machine learning has been primarily operating in the closed-world setting, where the classes are assumed the same between training and test data. Formally, let $X = \mathbb{R}^d$ denote the input space and $Y_{\text{in}} = \{1, \ldots, C\}$ denote the label space, with $C$ being the number of classes. The learner has access to the labeled training set $\mathcal{D}_{\text{in}} = \{(\mathbf{x}_i, y_i)\}_{i=1}^{n}$, drawn *i.i.d.* from the joint data distribution $\mathbb{P}_{XY_{\text{in}}}$. Let $\mathbb{P}_{\text{in}}$ denote the marginal distribution on $X$,

which is also referred to as the *in-distribution*. Let $f : X \mapsto \mathbb{R}^C$ denote a function for the classification task, which predicts the label of an input sample. To obtain an optimal classifier, a classic approach is called empirical risk minimization (ERM) (Vapnik, 1999): $f^* = \mathrm{argmin}_{f \in \mathcal{F}} R_{\mathrm{closed}}(f)$, where $R_{\mathrm{closed}}(f) = \frac{1}{n} \sum_{i=1}^n \ell(f(\mathbf{x}_i), y_i)$ and $\ell$ is the loss function and $\mathcal{F}$ is the hypothesis space.

**Out-of-distribution detection.** The closed-world assumption rarely holds for models deployed in the open world, where data from unknown classes can naturally emerge (Bendale & Boult, 2015). Formally, our framework concerns a common real-world scenario in which the algorithm is trained on the ID data with classes $Y_{\mathrm{in}} = \{1, ..., C\}$, but will then be deployed in environments containing *out-of-distribution* samples from unknown class $y \notin Y_{\mathrm{in}}$ and therefore should not be predicted by $f$. At its core, OOD detection estimates the lower level set $\mathcal{L} := \{\mathbf{x} : \mathbb{P}_{\mathrm{in}}(\mathbf{x}) \leq \beta\}$. Given any $\mathbf{x}$, it classifies $\mathbf{x}$ as OOD if and only if $\mathbf{x} \in \mathcal{L}$. The threshold $\beta$ is chosen by controlling the false detection rate: $\int_{\mathcal{L}} \mathbb{P}_{\mathrm{in}}(\mathbf{x})d\mathbf{x} = 0.05$. The false rate can be chosen as a typical (*e.g.*, 0.05) or another value appropriate for the application. As in classic statistics (see (Chen et al., 2017) and the references therein), we estimate the lower level set $\mathcal{L}$ from the ID dataset $\mathcal{D}_{\mathrm{in}}$.

# 3 METHOD

**Framework overview.** Machine learning models deployed in the wild must operate with *both* classification accuracy and safety performance. We use safety to characterize the model's ability to detect OOD data. This safety performance is lacking for off-the-shelf machine learning algorithms — which typically focus on minimizing error *only* on the in-distribution data from $\mathbb{P}_{\mathrm{in}}$, but does not account for the uncertainty that could arise outside $\mathbb{P}_{\mathrm{in}}$. For example, recall that empirical risk minimization (ERM) (Vapnik, 1999), a long-established method that is commonly used today, operates under the closed-world assumption (*i.e.*, no distribution shift between training vs. testing). Models optimized with ERM are known to produce overconfidence predictions on OOD data (Nguyen et al., 2015), since the decision boundary is not conservative.

To address the challenges, our learning framework jointly optimizes for both: **(1)** accurate classification of samples from $\mathbb{P}_{\mathrm{in}}$, and **(2)** reliable detection of data from outside $\mathbb{P}_{\mathrm{in}}$. Given a weighting factor $\alpha$, the risk can be formalized as follows:

$$\mathrm{argmin} \ [ \ \underbrace{R_{\mathrm{closed}}(f)}_{\text{Classification error on ID}} \ + \ \alpha \cdot \ \underbrace{R_{\mathrm{open}}(g)}_{\text{Error of OOD detector}} \ ], \tag{1}$$

where $R_{\mathrm{closed}}(f)$ aims to classify ID samples into known classes, and $R_{\mathrm{open}}(g)$ aims to distinguish ID vs. OOD. This framework thus integrates distributional uncertainty as a first-class construct. Our newly introduced risk term $R_{\mathrm{open}}(g)$ is crucial to prevent overconfident predictions for unknown data, and to improve test-time detection of unknowns. In the sequel, we introduce the two risks $R_{\mathrm{open}}(g)$ (Section 3.1) and $R_{\mathrm{closed}}(f)$ (Section 3.2) in detail, with emphasis placed on the former.

## 3.1 FORMALIZE $R_{\mathrm{OPEN}}(g)$

**OOD detection via level-set estimation.** To formalize $R_{\mathrm{open}}(g)$, we can explicitly perform a $\beta$-level set estimation, *i.e.*, binary classification in realization, between ID and OOD data. Concretely, the decision boundary is the empirical $\beta$ level set $\{\mathbf{x} : \hat{\mathbb{P}}_{\mathrm{in}}(\mathbf{x}) = \beta\}$. Consider the following OOD conditional distribution:

$$Q(\mathbf{x} \mid \mathrm{OOD}) = \frac{\mathbf{1}[\hat{\mathbb{P}}_{\mathrm{in}}(\mathbf{x}) \leq \beta]\hat{\mathbb{P}}_{\mathrm{in}}(\mathbf{x})}{\mathcal{Z}_{\mathrm{out}}}, \quad \mathcal{Z}_{\mathrm{out}} = \int \mathbf{1}[\hat{\mathbb{P}}_{\mathrm{in}}(\mathbf{x}) \leq \beta]\hat{\mathbb{P}}_{\mathrm{in}}(\mathbf{x})d\mathbf{x}. \tag{2}$$

where $\mathbf{1}[\cdot]$ is the indicator function. $Q(\mathbf{x} \mid \mathrm{OOD})$ is $\hat{\mathbb{P}}_{\mathrm{in}}$ restricted to $\hat{\mathcal{L}} := \{\mathbf{x} : \hat{\mathbb{P}}_{\mathrm{in}}(\mathbf{x}) \leq \beta\}$ and renormalized. Similarly, define an ID conditional distribution:

$$Q(\mathbf{x} \mid \mathrm{ID}) = \frac{\mathbf{1}[\hat{\mathbb{P}}_{\mathrm{in}}(\mathbf{x}) > \beta]\hat{\mathbb{P}}_{\mathrm{in}}(\mathbf{x})}{1 - \mathcal{Z}_{\mathrm{out}}}. \tag{3}$$

Note $Q(\mathbf{x} \mid \mathrm{OOD})$ and $Q(\mathbf{x} \mid \mathrm{ID})$ have disjoint support that partition $h(X)$, where $h : X \mapsto \mathbb{R}^d$ is a feature encoder, which maps an input to the penultimate layer with $d$ dimensions. In particular,

for any non-degenerate prior $Q(\text{OOD}) = 1 - Q(\text{ID})$, the Bayes decision boundary for the joint distribution $Q$ is precisely the empirical $\beta$ level set.

Since we only have access to the ID training data, a critical consideration is *how to provide OOD data for training*. A recent work by Du et al. (2022c) proposed synthesizing virtual outliers from the low-likelihood region in the feature space $h(\mathbf{x})$, which is more tractable than synthesizing $\mathbf{x}$ in the input space $X$. However, they modeled the feature space as class-conditional Gaussian distribution — a strong assumption that might not always hold in practice. To circumvent this limitation, our new idea is to perform *non-parametric outlier synthesis*, which does not make any distributional assumption on the ID embeddings. Our proposed synthesis approach thereby offers stronger flexibility and generality.

To synthesize outlier data, we formalize our idea by rejection sampling (Rubinstein & Kroese, 2016) with $\hat{\mathbb{P}}_{\text{in}}$ as the proposal distribution. In a nutshell, the rejection sampling can be done in three steps:

1. Draw an index in $\mathcal{D}_{\text{in}}$ by $i \sim \text{uniform}[n]$ where $n$ is the number of training samples.

2. Draw sample $\mathbf{v}$ (candidate synthesized outlier) in the feature space from a Gaussian kernel centered at $h(\mathbf{x}_i)$ with covariance $\sigma^2 \mathbf{I}$: $\mathbf{v} \sim \mathcal{N}(h(\mathbf{x}_i), \sigma^2 \mathbf{I})$.

3. Accept $\mathbf{v}$ with probability $\frac{Q(h(\mathbf{x})|\text{OOD})}{M\hat{\mathbb{P}}_{\text{in}}}$ where $M$ is an upper bound on the likelihood ratio $Q(h(\mathbf{x}) \mid \text{OOD})/\hat{\mathbb{P}}_{\text{in}}$. Since $Q(h(\mathbf{x}) \mid \text{OOD})$ is truncated $\hat{\mathbb{P}}_{\text{in}}$, one can choose $M = 1/\mathcal{Z}_{\text{out}}$. Equivalently, accept $\mathbf{v}$ if $\hat{\mathbb{P}}_{\text{in}}(\mathbf{v}) < \beta$.

Despite the soundness of the mathematical framework, the realization in modern neural networks is non-trivial. A salient challenge is the computational efficiency — drawing samples uniformly from $\hat{\mathbb{P}}_{\text{in}}$ in step 1 is expensive since the majority of samples will have a high density that is easily rejected by step 3. To realize our framework efficiently, we propose the following procedures: (1) identify boundary ID samples, and (2) synthesize outliers based on the boundary samples.

**Identify ID samples near the boundary.** We leverage the non-parametric nearest neighbor distance as a heuristic surrogate for approximating $\hat{\mathbb{P}}_{\text{in}}$, and select the ID data with the highest $k$-NN distances as the boundary samples. We illustrate this step in the middle panel of Figure 1. Specifically, denote the embedding set of training data as $\mathbb{Z} = (\mathbf{z}_1, \mathbf{z}_2, ..., \mathbf{z}_n)$, where $\mathbf{z}_i$ is the $L_2$-normalized penultimate feature $\mathbf{z}_i = h(\mathbf{x}_i)/\|h(\mathbf{x}_i)\|_2$. For any embedding $\mathbf{z}' \in \mathbb{Z}$, we calculate the $k$-NN distance *w.r.t.* $\mathbb{Z}$:

$$d_k(\mathbf{z}', \mathbb{Z}) = \|\mathbf{z}' - \mathbf{z}_{(k)}\|_2, \tag{4}$$

where $\mathbf{z}_{(k)}$ is the $k$-th nearest neighbor in $\mathbb{Z}$. If an embedding has a large $k$-NN distance, it is likely to be on the boundary in the feature space. Thus, according to the $k$-NN distance, we select embeddings with the largest $k$-NN distances. We denote the set of boundary samples as $\mathbb{B}$.

**Synthesize outliers based on boundary samples.** Now we have obtained a set of ID embeddings near the boundary in the feature space, we synthesize outliers by sampling from a multivariate Gaussian distribution centered around the selected ID embedding $h(\mathbf{x}_i) \in \mathbb{B}$:

$$\mathbf{v} \sim \mathcal{N}(h(\mathbf{x}_i), \sigma^2 \mathbf{I}), \tag{5}$$

where the $\mathbf{v}$ denotes the synthesized outliers around $h(\mathbf{x}_i)$, and $\sigma^2$ modulates the variance. For each boundary ID sample, we can repeatedly sample $p$ different outliers using Equation 5, which produces a set $V_i = (\mathbf{v}_1, \mathbf{v}_2, ..., \mathbf{v}_p)$. To ensure that the outliers are sufficiently far away from the ID data, we further perform a filtering process by selecting the virtual outlier in $V_i$ with the highest $k$-NN distance *w.r.t.* $\mathbb{Z}$, as illustrated in the right panel of Figure 1 (dark orange points).

The final collection of accepted virtual outliers will be used for the binary training objective:

$$R_{\text{open}} = \mathbb{E}_{\mathbf{v} \sim \mathcal{V}} \left[ -\log \frac{1}{1 + \exp^{\phi(\mathbf{v})}} \right] + \mathbb{E}_{\mathbf{x} \sim \mathbb{P}_{\text{in}}} \left[ -\log \frac{\exp^{\phi(h(\mathbf{x}))}}{1 + \exp^{\phi(h(\mathbf{x}))}} \right], \tag{6}$$

where $\phi(\cdot)$ is a nonlinear MLP function and $h(\mathbf{x})$ denotes the ID embeddings. In other words, the loss function takes both the ID and synthesized outlier embeddings and aims to estimate the level set through the binary cross-entropy loss.

Table 1: OOD detection performance on ImageNet-100 (Deng et al., 2009) as ID. All methods are trained on the same backbone. Values are percentages. **Bold** numbers are superior results. ↑ indicates larger values are better, and ↓ indicates smaller values are better.

| Methods | OOD Datasets | | | | | | | | | | ID ACC↑ |
| | iNaturalist | | SUN | | Places | | Textures | | Average | | |
| | FPR95↓ | AUROC↑ | FPR95↓ | AUROC↑ | FPR95↓ | AUROC↑ | FPR95↓ | AUROC↑ | FPR95↓ | AUROC↑ | |
|---|---|---|---|---|---|---|---|---|---|---|---|
| MCM (zero-shot) | 15.23 | 97.30 | 25.05 | 95.95 | 24.91 | 95.66 | 33.68 | 94.11 | 24.72 | 95.76 | 89.32 |
| *(Fine-tuned)* | | | | | | | | | | | |
| Fort et al./MSP | 49.48 | 93.72 | 38.56 | 93.16 | 41.30 | 91.53 | 38.16 | 93.53 | 41.87 | 92.98 | 94.64 |
| ODIN | 7.32 | 98.09 | 29.05 | 92.63 | 32.58 | 92.60 | 15.50 | 96.58 | 21.11 | 94.98 | 94.64 |
| Energy | 20.95 | 95.94 | 18.42 | 94.77 | 22.35 | 93.08 | 15.36 | 96.19 | 19.27 | 94.99 | 94.64 |
| GradNorm | 28.72 | 91.14 | 54.41 | 73.08 | 43.02 | 81.13 | 28.28 | 91.29 | 38.61 | 84.16 | 94.64 |
| ViM | 13.75 | 96.67 | 31.68 | 93.31 | 39.61 | 88.95 | 26.94 | 94.13 | 28.00 | 93.27 | 94.64 |
| KNN | 6.77 | 98.21 | 26.72 | 92.87 | 23.13 | 92.81 | 15.02 | 96.73 | 17.91 | 95.16 | 94.64 |
| VOS | 8.05 | 98.03 | 29.54 | 92.91 | 29.54 | 92.91 | 13.79 | 97.28 | 19.16 | 95.69 | 94.14 |
| VOS+ | 8.44 | 98.00 | 17.58 | 96.69 | 17.46 | 95.95 | 16.85 | 96.43 | 15.08 | 96.77 | 94.38 |
| **NPOS** (ours) | **0.70** | **99.14** | **9.22** | **98.48** | **5.12** | **98.86** | **8.01** | **98.47** | **5.76** | **98.74** | 94.76 |

## 3.2 OPTIMIZING ID EMBEDDINGS WITH $R_{\text{CLOSED}}(f)$

Now we discuss the design of $R_{\text{closed}}$, which minimizes the risk on the in-distribution data. We aim to produce highly distinguishable ID representations, which non-parametric outlier synthesis (*cf.* Section 3.1) depends on. In a nutshell, the model aims to learn compact representations that align ID samples with their class prototypes. Specifically, we denote by $\boldsymbol{\mu}_1, \boldsymbol{\mu}_2, \ldots, \boldsymbol{\mu}_C$ the prototype embeddings for the ID classes $c \in \{1, 2, .., C\}$. The prototype for each sample is assigned based on the ground-truth class label. For any input $\mathbf{x}$ with corresponding embedding $h(\mathbf{x})$, we can calculate the cosine similarity between $h(\mathbf{x})$ and prototype vector $\boldsymbol{\mu}_j$:

$$f_j(\mathbf{x}) = \frac{h(\mathbf{x}) \cdot \boldsymbol{\mu}_j}{\|h(\mathbf{x})\| \cdot \|\boldsymbol{\mu}_j\|}, \tag{7}$$

which can be viewed as the $j$-th logit output. A larger $f_j(\mathbf{x})$ indicates a stronger association with the $j$-th class. The classification loss is the cross-entropy applied to the softmax output:

$$R_{\text{closed}} = -\log\left(\frac{e^{f_y(\mathbf{x})/(\tau \cdot \|f\|)}}{\sum_{j=1}^{C} e^{f_j(\mathbf{x})/(\tau \cdot \|f\|)}}\right), \tag{8}$$

where $\tau$ is the temperature, and $f_y(\mathbf{x})$ is the logit output corresponding to the ground truth label $y$.

Our training framework is end-to-end trainable, where the two losses $R_{\text{open}}$ (*cf.* Section 3.1) and $R_{\text{closed}}$ work in a synergistic fashion. First, as the classification loss (Equation 8) shapes ID embeddings, our non-parametric outlier synthesis module benefits from the highly distinguishable representations. Second, our uncertainty loss in Equation 6 would facilitate learning a compact decision boundary between ID and OOD, which provides a reliable estimation for OOD uncertainty that can arise. The entire training process converges when the two components perform satisfactorily.

## 3.3 TEST-TIME OOD DETECTION

In testing, we use the same scoring function as Ming et al. (2022a) for OOD detection $S(\mathbf{x}) = \max_j \frac{e^{f_j(\mathbf{x})/\tau}}{\sum_{c=1}^{C} e^{f_c(\mathbf{x})/\tau}}$, where $f_j(\mathbf{x}) = \frac{h(\mathbf{x}) \cdot \boldsymbol{\mu}_j}{\|h(\mathbf{x})\| \cdot \|\boldsymbol{\mu}_j\|}$. The rationale is that, for ID data, it will be matched to one of the prototype vectors with a high score and vice versa. Based on the scoring function, the OOD detector is $G_\lambda(\mathbf{x}) = \mathbf{1}\{S(\mathbf{x}) \geq \lambda\}$, where by convention, 1 represents the positive class (ID), and 0 indicates OOD. $\lambda$ is chosen so that a high fraction of ID data (*e.g.*, 95%) is above the threshold. Our algorithm is summarized in Algorithm 1 (Appendix C).

## 4 EXPERIMENTS

In this section, we present empirical evidence to validate the effectiveness of our method on real-world classification tasks. We describe the setup in Section 4.1, followed by the results and comprehensive analysis in Section 4.2–Section 4.5.

## 4.1 SETUP

**Datasets.** We use both standard CIFAR-100 benchmark (Krizhevsky et al., 2009) and the large-scale ImageNet dataset (Deng et al., 2009) as the in-distribution data. Our main results and ablation

Table 2: OOD detection performance for ImageNet-1k (Deng et al., 2009) as ID.

| Methods | OOD Datasets | | | | | | | | | | ID ACC ↑ |
| | iNaturalist | | SUN | | Places | | Textures | | Average | | |
| | FPR95↓ | AUROC↑ | FPR95↓ | AUROC↑ | FPR95↓ | AUROC↑ | FPR95↓ | AUROC↑ | FPR95↓ | AUROC↑ | |
|---|---|---|---|---|---|---|---|---|---|---|---|
| MCM (zero-shot) | 32.08 | 94.41 | 39.21 | 92.28 | 44.88 | 89.83 | 58.05 | 85.96 | 43.55 | 90.62 | 68.53 |
| *(Fine-tuned)* | | | | | | | | | | | |
| Fort et al./MSP | 54.05 | 87.43 | 73.37 | 78.03 | 72.98 | 78.03 | 68.85 | 79.06 | 67.31 | 80.64 | 79.64 |
| ODIN | 30.22 | 94.65 | 54.04 | 87.17 | 55.06 | 85.54 | 51.67 | 87.85 | 47.75 | 88.80 | 79.64 |
| Energy | 29.75 | 94.68 | 53.18 | 87.33 | 56.40 | 85.60 | 51.35 | 88.00 | 47.67 | 88.90 | 79.64 |
| GradNorm | 81.50 | 72.56 | 82.00 | 72.86 | 80.41 | 73.70 | 79.36 | 70.26 | 80.82 | 72.35 | 79.64 |
| ViM | 32.19 | 93.16 | 54.01 | 87.19 | 60.67 | 83.75 | 53.94 | 87.18 | 50.20 | 87.82 | 79.64 |
| KNN | 29.17 | 94.52 | **35.62** | **92.67** | 39.61 | 91.02 | 64.35 | 85.67 | 42.19 | 90.97 | 79.64 |
| VOS | 31.65 | 94.53 | 43.03 | 91.92 | 41.62 | 90.23 | 56.67 | 86.74 | 43.24 | 90.86 | 79.64 |
| VOS+ | 28.99 | 94.62 | 36.88 | 92.57 | **38.39** | **91.23** | 61.02 | 86.33 | 41.32 | 91.19 | 79.58 |
| **NPOS** (ours) | **16.58** | **96.19** | 43.77 | 90.44 | 45.27 | 89.44 | **46.12** | **88.80** | **37.93** | **91.22** | 79.42 |

studies are based on ImageNet-100, which consists of 100 randomly selected classes from the original ImageNet-1k dataset. For completeness, we also conduct experiments on the full ImageNet-1k dataset (Section 4.2). For OOD datasets, we adopt the same ones as in (Huang & Li, 2021), including subsets of iNaturalist (Van Horn et al., 2018), SUN (Xiao et al., 2010), PLACES (Zhou et al., 2017), and TEXTURE (Cimpoi et al., 2014). For each OOD dataset, the categories are disjoint from the ID dataset. We provide details of the datasets and categories in Appendix A.

**Model.** In our main experiments, we perform training by fine-tuning the CLIP model (Radford et al., 2021), which is one of the most popular and publicly available pre-trained models. CLIP aligns an image with its corresponding textual description in the feature space by a self-supervised contrastive objective. Concretely, it adopts a simple dual-stream architecture with one image encoder $\mathcal{I} : \mathbf{x} \to \mathbb{R}^d$ (*e.g.*, ViT (Dosovitskiy et al., 2021)), and one text encoder $\mathcal{T} : t \to \mathbb{R}^d$ (*e.g.*, Transformer (Vaswani et al., 2017)). We fine-tune the last two blocks of CLIP's image encoder, using our proposed training objective in Section 3. To indicate input patch size in ViT models, we append "/x" to model names. We prepend -B, -L to indicate Base and Large versions of the corresponding architecture. For instance, ViT-B/16 implies the Base variant with an input patch resolution of $16 \times 16$. We mainly use CLIP-B/16, which contains a ViT-B/16 Transformer as the image encoder. We utilize the pre-extracted text embeddings from a masked self-attention Transformer as the prototypes for each class, where $\boldsymbol{\mu}_i = \mathcal{T}(t_i)$ and $t_i$ is the text prompt for a label $y_i$. The text encoder is not needed in the training process. We additionally conduct ablations on alternative backbone architecture including ResNet in Section 4.3. Note that our method is not limited to pre-trained models; it can generally be applicable for models trained from scratch, as we will later show in Section 4.4.

**Experimental details.** We employ a two-layer MLP with a ReLU nonlinearity for $\phi$, with a hidden layer dimension of 16. We train the model using stochastic gradient descent with a momentum of 0.9, and weight decay of $10^{-4}$. For ImageNet-100, we train the model for a total of 20 epochs, where we only use Equation 8 for representation learning for the first ten epochs. We train the model jointly with our outlier synthesis loss (Equation 6) in the last 10 epochs. We set the learning rate to be 0.1 for the $R_{\text{closed}}$ branch, and 0.01 for the MLP in the $R_{\text{open}}$ branch. For the ImageNet-1k dataset, we train the model for 60 epochs, where the first 20 epochs are trained with Equation 8. Extensive ablations on the hyperparameters are conducted in Section 4.3 and Appendix F.

**Evaluation metrics.** We report the following metrics: (1) the false positive rate (FPR95) of OOD samples when the true positive rate of ID samples is 95%, (2) the area under the receiver operating characteristic curve (AUROC), and (3) ID classification error rate (ID ERR).

## 4.2 MAIN RESULTS

**NPOS significantly improves OOD detection performance.** As shown in the Table 1, we compare the proposed NPOS with competitive OOD detection methods. For a fair comparison, all the methods only use ID data without using auxiliary outlier datasets. We compare our methods with the following recent competitive baselines, including (1) Maximum Concept Matching (MCM) (Ming et al., 2022a), (2) Maximum Softmax Probability (Hendrycks & Gimpel, 2017; Fort et al., 2021), (3) ODIN score (Liang et al., 2018), (4) Energy score (Liu et al., 2020b), (5) GradNorm score (Huang et al., 2021), (6) ViM score (Wang et al., 2022), (7) KNN distance (Sun et al., 2022), and (8) VOS

Table 3: Ablation on model capacity and architecture. ID dataset is ImageNet-100.

| Model | Methods | iNaturalist | | SUN | | Places | | Textures | | Average | | ID ACC |
|---|---|---|---|---|---|---|---|---|---|---|---|---|
| | | FPR95 | AUROC | FPR95 | AUROC | FPR95 | AUROC | FPR95 | AUROC | FPR95 | AUROC | |
| **RN50x4** | Fort et al. | 66.14 | 87.19 | 61.90 | 87.34 | 60.50 | 87.16 | 47.54 | 90.51 | 59.02 | 88.05 | 92.26 |
| | MCM | 20.15 | 96.61 | 22.52 | 95.92 | 27.04 | 94.54 | 28.39 | 93.97 | 24.54 | 95.26 | 87.84 |
| | Ours | **3.12** | **99.15** | **4.75** | **98.30** | **12.87** | **98.75** | **11.02** | **97.93** | **7.94** | **98.53** | 92.32 |
| **ViT-B/16** | Fort et al. | 49.48 | 93.72 | 38.56 | 93.16 | 41.30 | 91.53 | 38.16 | 93.53 | 41.87 | 92.98 | 94.64 |
| | MCM | 15.23 | 97.30 | 25.05 | 95.95 | 24.91 | 95.66 | 33.68 | 94.11 | 24.72 | 95.76 | 89.32 |
| | Ours | **0.70** | **99.14** | **9.22** | **98.48** | **5.12** | **98.86** | **8.01** | **98.47** | **5.76** | **98.74** | 94.76 |
| **ViT-L/14** | Fort et al. | 26.63 | 93.34 | 34.75 | 93.62 | 33.75 | 92.13 | 32.95 | 91.95 | 32.02 | 92.76 | 96.39 |
| | MCM | 15.09 | 97.49 | 14.25 | 97.42 | 16.72 | 96.75 | 27.02 | 94.34 | 18.27 | 96.50 | 92.30 |
| | Ours | **0.13** | **99.43** | **4.79** | **99.10** | **6.36** | **98.42** | **7.63** | **98.19** | **4.73** | **98.79** | 96.28 |

(Du et al., 2022b) which synthesizes outliers by modeling the ID embeddings as a mixture of Gaussian distribution and sampling from the low-likelihood region of the feature space. MCM is the latest zero-shot OOD detection approach for vision-language models. All the other baseline methods are fine-tuned using the same pre-trained model weights (*i.e.*, CLIP-B/16) and the same number of layers as ours. We added a fully connected layer to the model backbone, which produces the classification output. In particular, Fort et al. (2021) fine-tuned the model using cross-entropy loss and then applied the MSP score in testing. KNN distance is calculated using features from the penultimate layer of the same fine-tuned model as Fort et al. (2021). For VOS, we follow the original loss function and OOD score defined in Du et al. (2022b).

We show that NPOS can achieve superior OOD detection performance, outperforming the competitive rivals by a large margin. In particular, NPOS reduces the FPR95 from 41.87% (Fort et al. (2021)) to 5.76% (ours) — a direct **36.11%** improvement. The performance gap signifies the effectiveness of our training loss using outlier synthesis for model regularization. By incorporating the synthesized outliers, our risk term $R_{\mathrm{open}}$ is crucial to prevent overconfident predictions for OOD data, and improve test-time OOD detection.

**Non-parametric outlier synthesis outperforms VOS.** We contrast NPOS with the most relevant baseline VOS, where NPOS outperforms by 13.40% in FPR95. A major difference between the two approaches lies in how outliers are synthesized: parametric approach (VOS) vs. non-parametric approach (ours). Compared to VOS, our method does not make any distributional assumption on the ID embeddings, hence offering stronger flexibility and generality. Another difference lies in the ID classification loss: VOS employs the softmax cross-entropy loss while our method utilizes a different loss (*cf.* Equation 8) to learn distinguishable ID embeddings. To clearly isolate the effect, we further enhance VOS by using the same classification loss as defined in Equation 8 and endow VOS with a stronger representation space. This resulting method dubbed VOS+ and its corresponding performance is also shown in Table 1. Note that VOS+ and NPOS *only* differ in how outliers are synthesized. While VOS+ indeed performs better compared to the original VOS, the FPR95 is still worse than our method. This experiment directly and fairly confirms the superiority of our proposed non-parametric outlier synthesis approach. Computationally, the training time using NPOS is 30.8 minutes on ImageNet-100, which is comparable to VOS (30.0 minutes).

**NPOS scales effectively to large datasets.** To examine the scalability of NPOS, we also evaluate on the ImageNet-1k dataset (ID) in Table 2. Recently, Fort et al. (2021) explored small-scale OOD detection by fine-tuning the ViT model, and then applying the MSP score in testing. When extending to large-scale tasks, we find that NPOS yields superior performance under the same image encoder configuration (ViT-B/16). In particular, NPOS reduces the FPR95 from 67.31% to 37.93%. This highlights the advantage of utilizing non-parametric outlier synthesis to learn a conservative decision boundary for OOD detection. Our results also confirm that NPOS can indeed scale to large datasets with complex visual diversity.

**Learning compact ID representation benefits NPOS.** We investigate the importance of optimizing ID embeddings for non-parametric outlier synthesis. Recall that our loss function in Equation 8 facilitates learning a compact representation for ID data. To isolate its effect, we replace the loss function with the cross-entropy loss while keeping everything else the same. On ImageNet-100, this yields an average FPR95 of 17.94%, and AUROC 95.75%. The worsened OOD detection performance signifies the importance of our ID classification loss that optimizes ID embedding quality.

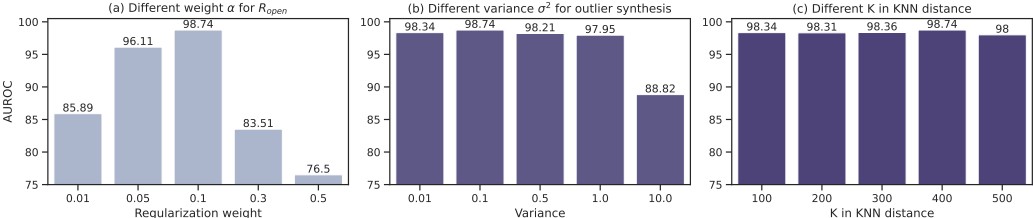

Figure 2: (a) Ablation study on the regularization weight $\alpha$ on $R_{\text{open}}$. (b) Ablation on the variance $\sigma^2$ for synthesizing outliers in Equation 5. (c) Ablation on the $k$ for the $k$-NN distance. The numbers are FPR95. The ID training dataset is ImageNet-100.

### 4.3 ABLATION STUDY

**Ablation on model capacity and architecture.** To show the effectiveness of the ResNet-based architecture, we replace the CLIP image encoder with RN50x4 (178.3M), which shares a similar number of parameters as CLIP-B/16 (149.6M). The OOD detection performance of NPOS for the ImageNet-100 dataset (ID) is shown in Figure 3. It can be seen that NPOS still shows promising results with the ResNet-based backbone, and the performance is comparable between RN50x4 and CLIP-B/16 (7.94 vs. 5.76 in FPR95). Moreover, we observe that a larger model capacity indeed leads to stronger performance. Compared with CLIP-B, NPOS based on CLIP-L/14 reduces FPR95 to 4.73%. This suggests that larger models are endowed with a better representation quality which benefits outlier synthesis and OOD detection. Our findings corroborate the observations in Du et al. (2022c) that a higher-capacity model is correlated with stronger OOD detection performance.

**Ablation on the loss weight.** Recall that our training objective consists of two risks $R_{\text{closed}}$ and $R_{\text{open}}$. The two losses are combined with a weight $\alpha$ on $R_{\text{open}}$, and constant 1 on $R_{\text{closed}}$. In Figure 2 (a), we ablate the effect of weight $\alpha$ on the OOD detection performance. Here the ID data is ImageNet-100. When $\alpha$ reduces to a small value (*e.g.*, 0.01), the performance becomes closer to the MSP baseline trained with $R_{\text{closed}}$ only. In contrast, under a mild weighting, such as $\alpha = 0.1$, the OOD detection performance is significantly improved. Too excessive regularization using synthesized outliers ultimately degrades the performance.

**Ablation on the variance $\sigma^2$ in sampling.** A proper variance $\sigma^2$ in sampling virtual outliers is critical to our method. Recall in Equation 5 that $\sigma^2$ modulates the variance when synthesizing outliers around boundary samples. In Figure 2 (b), we systematically analyze the effect of $\sigma^2$ on OOD detection performance. We vary $\sigma^2 = \{0.01, 0.1, 0.5, 1.0, 10\}$. We observe that the performance of NPOS is *insensitive* under a moderate variance. In the extreme case when $\sigma^2$ becomes too large, the sampled virtual outliers might suffer from severe overlapping with ID data, which leads to performance degradation as expected.

**Ablation on $k$ in calculating KNN distance.** In Figure 2 (c), we analyze the effect of $k$, *i.e.*, the number of nearest neighbors for non-parametric density estimation. In particular, we vary $k = \{100, 200, 300, 400, 500\}$. We observe that our method is not sensitive to this hyperparameter, as $k$ varies from 100 to 500.

### 4.4 ADDITIONAL RESULTS WITHOUT PRE-TRAINED MODEL

Going beyond fine-tuning with the large pre-trained model, we show that NPOS is also applicable and effective when training from scratch. This setting allows us to evaluate our algorithm itself without the impact of strong model initialization. Appendix D showcases the performance of NPOS trained on three datasets: CIFAR-10, CIFAR-100, and ImageNet-100 datasets. We substitute the text embeddings $\boldsymbol{\mu}_i$ in vision-language models with the class-conditional image embeddings estimated in an exponential-moving-average (EMA) manner (Li et al., 2021): $\boldsymbol{\mu}_c := \text{Normalize}(\gamma\boldsymbol{\mu}_c + (1 - \gamma)\mathbf{z})$, $\forall c \in \{1, 2, \ldots, C\}$, where the prototype $\boldsymbol{\mu}_c$ for class $c$ is updated during training as the moving average of all embeddings with label $c$, and $\mathbf{z}$ denotes the normalized embedding of samples in class $c$. The EMA style update avoids the costly alternating training and prototype estimation over the entire training set as in the conventional approach (Zhe et al., 2019).

We evaluate on six OOD datasets: TEXTURES (Cimpoi et al., 2014), SVHN (Netzer et al., 2011), PLACES365 (Zhou et al., 2017), LSUN-RESIZE & LSUN-C (Yu et al., 2015), and ISUN (Xu et al., 2015). The comparison is shown in Table 4, Table 5, and Table 6. NPOS consistently improves OOD detection performance over all the published baselines. For example, on CIFAR-100, NPOS outperforms the most relevant baseline VOS by 27.41% in FPR95.

### 4.5 VISUALIZATION ON THE SYNTHESIZED OUTLIERS

Qualitatively, we show the t-SNE visualization (Van der Maaten & Hinton, 2008) of the synthesized outliers by our proposed method NPOS in Figure 3. The ID features (colored in purple) are extracted from the penultimate layer of a model trained on ImageNet-100 (class name: HERMIT CRAB). Without making any distributional assumption on the embedding space, NPOS is able to synthesize outliers (colored in orange) in the low-likelihood region, thereby offering strong flexibility and generality.

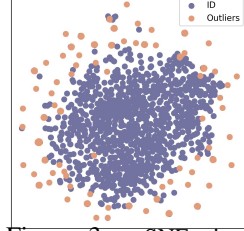

Figure 3: t-SNE visualization of synthesized outliers by NPOS.

## 5 RELATED WORK

**OOD detection** has attracted a surge of interest since the overconfidence phenomenon in OOD data is first revealed in Nguyen et al. (2015). One line of work performs OOD detection by devising scoring functions, including confidence-based methods (Bendale & Boult, 2016; Hendrycks & Gimpel, 2017; Liang et al., 2018), energy-based score (Liu et al., 2020b; Wang et al., 2021), distance-based approaches (Lee et al., 2018b; Tack et al., 2020; Sehwag et al., 2021; Sun et al., 2022; Du et al., 2022a; Ming et al., 2023), gradient-based score (Huang et al., 2021), and Bayesian approaches (Gal & Ghahramani, 2016; Lakshminarayanan et al., 2017; Maddox et al., 2019; Malinin & Gales, 2019; Wen et al., 2020; Kristiadi et al., 2020).

Another promising line of work addressed OOD detection by training-time regularization (Bevandić et al., 2018; Malinin & Gales, 2018; Geifman & El-Yaniv, 2019; Hein et al., 2019; Meinke & Hein, 2020; Mohseni et al., 2020; Jeong & Kim, 2020; van Amersfoort et al., 2020; Yang et al., 2021; Wei et al., 2022). For example, the model is regularized to produce lower confidence (Lee et al., 2018a; Hendrycks et al., 2019) or higher energy (Du et al., 2022c; Liu et al., 2020b; Katz-Samuels et al., 2022; Ming et al., 2022b) on the outlier data. Most regularization methods require the availability of auxiliary OOD data. Among methods utilizing ID data only, Hsu et al. (2020) proposed to decompose confidence scoring during training with a modified input pre-processing method. Liu et al. (2020a) proposed a spectral-normalized neural Gaussian process by optimizing the network design for uncertainty estimation. Closest to our work, VOS (Du et al., 2022c) synthesizes virtual outliers using multivariate Gaussian distributions, and regularizes the model's decision boundary between ID and OOD data during training. In this paper, we propose a novel non-parametric outlier synthesis approach, mitigating the distributional assumption made in VOS.

**Large-scale OOD detection.** Recent works have advocated for OOD detection in large-scale settings, which are closer to real-world applications. Research efforts include scaling OOD detection to large semantic label space (Huang & Li, 2021) and exploiting large pre-trained models (Fort et al., 2021). Recently, powerful pre-trained vision-language models have achieved strong results on zero-shot OOD detection (Ming et al., 2022a). Different from prior works, we propose a new training/fine-tuning procedure with non-parametric outlier synthesis for model regularization. Our learning framework renders a conservative decision boundary between ID and OOD data, and thereby improves OOD detection.

## 6 CONCLUSION

In this paper, we propose a novel framework NPOS, which tackles ID classification and OOD uncertainty estimation in one coherent framework. NPOS mitigates the key shortcomings of the previous outlier synthesis-based OOD detection approach, and synthesizes outliers without imposing any distributional assumption. To the best of our knowledge, NPOS makes the first attempt to employ a non-parametric outlier synthesis for OOD detection and can be formally interpreted as a rejection sampling framework. NPOS establishes competitive performance on challenging real-world OOD detection tasks, evaluated broadly under both the recent vision-language models and models that are trained from scratch. Our in-depth ablations provide further insights on the efficacy of NPOS. We hope our work inspires future research on OOD detection based on non-parametric outlier synthesis.

## REPRODUCIBILITY STATEMENT

We summarize our efforts below to facilitate reproducible results:

1. **Datasets.** We use publicly available datasets, which are described in detail in Section 4.1, Section 4.4, and Appendix A.

2. **Baselines.** The description and hyperparameters of the OOD detection baselines are explained in Section 4.2, and Appendix B.

3. **Methodology.** Our method is fully documented in Section 3, with the pseudo algorithm detailed in Algorithm 1. Hyperparamters are specified in Section 4.1, with a thorough ablation study provided in Section 4.3 and Appendix F.

4. **Open Source.** Code, datasets and model checkpoints are publicly available at https://github.com/deeplearning-wisc/npos.

## ACKNOWLEDGEMENT

Li gratefully acknowledges the support of the AFOSR Young Investigator Award under No. FA9550-23-1-0184; Philanthropic fund from SFF; Wisconsin Alumni Research Foundation; faculty research awards from Google, Meta, and Amazon. Zhu is supported in part by NSF grants 1545481, 1704117, 1836978, 2023239, 2041428, 2202457, ARO MURI W911NF2110317, and AF CoE FA9550-18-1-0166. Any opinions, findings, conclusions or recommendations expressed in this material are those of the authors and do not necessarily reflect the views, policies, or endorsements, either expressed or implied of sponsors.

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

# Non-parametric Outlier Synthesis
# (Appendix)

## A  DETAILS OF DATASETS

**ImageNet-100.** We randomly sample 100 classes from ImageNet-1k (Deng et al., 2009) to create ImageNet-100. The dataset contains the following categories: n01986214, n04200800, n03680355, n03208938, n02963159, n03874293, n02058221, n04612504, n02841315, n02099712, n02093754, n03649909, n02114712, n03733281, n02319095, n01978455, n04127249, n07614500, n03595614, n04542943, n02391049, n04540053, n03483316, n03146219, n02091134, n02870880, n04479046, n03347037, n02090379, n10148035, n07717556, n04487081, n04192698, n02268853, n02883205, n02002556, n04273569, n02443114, n03544143, n03697007, n04557648, n02510455, n03633091, n02174001, n02077923, n03085013, n03888605, n02279972, n04311174, n01748264, n02837789, n07613480, n02113712, n02137549, n02111129, n01689811, n02099601, n02085620, n03786901, n04476259, n12998815, n04371774, n02814533, n02009229, n02500267, n04592741, n02119789, n02090622, n02132136, n02797295, n01740131, n02951358, n04141975, n02169497, n01774750, n02128757, n02097298, n02085782, n03476684, n03095699, n04326547, n02107142, n02641379, n04081281, n06596364, n03444034, n07745940, n03876231, n09421951, n02672831, n03467068, n01530575, n03388043, n03991062, n02777292, n03710193, n09256479, n02443484, n01728572, n03903868.

**OOD datasets.** Huang & Li curated a diverse collection of subsets from iNaturalist (Van Horn et al., 2018), SUN (Xiao et al., 2010), Places (Zhou et al., 2017), and Texture (Cimpoi et al., 2014) as large-scale OOD datasets for ImageNet-1k, where the classes of the test sets do not overlap with ImageNet-1k. We provide a brief introduction for each dataset as follows.

**iNaturalist** contains images of natural world (Van Horn et al., 2018). It has 13 super-categories and 5,089 sub-categories covering plants, insects, birds, mammals, and so on. We use the subset that contains 110 plant classes which are not overlapping with ImageNet-1k.

**SUN** stands for the Scene UNderstanding Dataset (Xiao et al., 2010). SUN contains 899 categories that cover more than indoor, urban, and natural places with or without human beings appearing in them. We use the subset which contains 50 natural objects not in ImageNet-1k.

**Places** is a large scene photographs dataset (Zhou et al., 2017). It contains photos that are labeled with scene semantic categories from three macro-classes: Indoor, Nature, and Urban. The subset we use contains 50 categories that are not present in ImageNet-1k.

**Texture** stands for the Describable Textures Dataset (Cimpoi et al., 2014). It contains images of textures and abstracted patterns. As no categories overlap with ImageNet-1k, we use the entire dataset as in Huang & Li (2021).

## B  BASELINES

To evaluate the baselines, we follow the original definition in MSP (Hendrycks & Gimpel, 2017; Fort et al., 2021), ODIN score (Liang et al., 2018), Energy score (Liu et al., 2020b), GradNorm score (Huang et al., 2021), ViM score (Wang et al., 2022), KNN distance (Sun et al., 2022) and VOS (Du et al., 2022b).

- For ODIN, we follow the original setting in the work and set the temperature $T$ as 1000.
- For both Energy and GradNorm scores, the temperature is set to be $T = 1$.
- For ViM, we follow the original implementation according to the released code.
- For VOS, we ensure that the number of negative samples is consistent with our method — for each class, we sample 60k points after estimating the distribution and select six outliers with the lowest likelihood. For the OOD score, we adopt the uncertainty proposed in the original method.
- For VOS+, we use the same loss function as defined in Section 3, but only replace the sampling method to be parametric. The way VOS+ synthesizes outliers is the same as VOS (first modeling the feature embedding as a mixture of multivariate Gaussian distribution, and then sample virtual outliers from the low-likelihood region in the embedding space). For a fair comparison, we also use the textual embedding extracted from CLIP as the prototype for VOS+. Note that VOS+ and NPOS *only* differs in how outliers are synthesized.

## C    ALGORITHM OF NPOS

We summarize our algorithm in implementation. Following (Du et al., 2022c), we construct a class-conditional in-distribution sample queue $\{Q_c\}_{c=1}^{C}$, which is periodically updated as new batches of training samples arrive.

---

**Algorithm 1** NPOS: Non-parametric Outlier Synthesis

---

**Input:** ID training data $\mathcal{D}_{\text{in}} = \{(\mathbf{x}_i, \mathbf{y}_i)\}_{i=1}^{n}$, initial model parameters $\theta$ for backbone, nonlinear MLP layer $\phi$ and class-conditional prototypes $\boldsymbol{\mu}$.
**Output:** Learned classifier $f(\mathbf{x})$, and OOD detector $G(\mathbf{x})$.
**while** *train* **do**

> 1. Update class-conditional queue $\{Q_c\}_{c=1}^{C}$ with the feature embeddings $h(\mathbf{x})$ of training samples in the current batch.
> 2. Select a set of boundary samples $\mathbb{B}_c$ consisting of top-$m$ embeddings with the largest $k$-NN distances using Equation 4.
> 3. Synthesize a set of outliers $V_i$ around each boundary sample $\mathbf{x}_i \in \mathbb{B}_1 \cup \mathbb{B}_2 \cup ...\mathbb{B}_C$ using Equation 5.
> 4. Accept the outliers in each $V_i$ with large $k$-NN distances.
> 5. Calculate level-set estimation loss $R_{\text{open}}$ and ID embedding optimization loss $R_{\text{closed}}$ using Equations 6 and 8, respectively, update the parameters $\theta, \phi$ based on loss in Equation 1.
> 6. Update prototypes using $\boldsymbol{\mu}_c := \text{Normalize}(\gamma\boldsymbol{\mu}_c + (1-\gamma)\mathbf{z}), \ \forall c \in \{1, 2, \ldots, C\}$.

**end**
**while** *eval* **do**

> 1. Calculate the OOD score defined in Section 3.3.
> 2. Perform OOD detection by thresholding comparison.

**end**

---

## D    EXPERIMENTAL DETAILS AND RESULTS ON TRAINING FROM SCRATCH

In this section, we provide the implementation details and the experimental results for NPOS trained from scratch. We evaluate on three datasets: CIFAR-10, CIFAR-100, and ImageNet-100. We summarize the training configurations of NPOS in Table 7.

**CIFAR-10 and CIFAR-100.**    The results on CIFAR-10 are shown in Table 4. All methods are trained on ResNet-18. We consider the same set of baselines as in the main paper. For the post-hoc OOD detection methods (MSP, ODIN, Energy score, GradNorm, ViM, KNN), we report the results by training the model with the cross-entropy loss for 100 epochs using stochastic gradient descent with momentum 0.9. The start learning rate is 0.1 and decays by a factor of 10 at epochs 50, 75, and 90 respectively. The batch size is set to 256. The average FPR95 of NPOS is 10.16%, significantly outperforming the best baseline VOS (27.88%). The results on CIFAR-100 are shown in Table 5, where the strong performance of NPOS holds.

Table 4: OOD detection performance on CIFAR-10 as ID. All methods are trained on ResNet-18. Values are percentages. **Bold** numbers are superior results.

| Methods | OOD Datasets | | | | | | | | | | | | ID ACC |
| | SVHN | | LSUN | | iSUN | | Texture | | Places365 | | Average | | |
| | FPR95 | AUROC | FPR95 | AUROC | FPR95 | AUROC | FPR95 | AUROC | FPR95 | AUROC | FPR95 | AUROC | |
| MSP | 59.66 | 91.25 | 45.21 | 93.80 | 54.57 | 92.12 | 66.45 | 88.50 | 62.46 | 88.64 | 57.67 | 90.86 | 94.21 |
| ODIN | 20.93 | 95.55 | 7.26 | **98.53** | 33.17 | 94.65 | 56.40 | 86.21 | 63.04 | 86.57 | 36.16 | 92.30 | 94.21 |
| Energy | 54.41 | 91.22 | 10.19 | 98.05 | 27.52 | **95.59** | 55.23 | 89.37 | 42.77 | 91.02 | 38.02 | 93.05 | 94.21 |
| GradNorm | 80.86 | 81.41 | 53.87 | 88.39 | 60.32 | 88.00 | 71.66 | 80.79 | 80.71 | 72.57 | 69.49 | 82.23 | 94.21 |
| ViM | 24.95 | 95.36 | 18.80 | 96.63 | 29.25 | 95.10 | 24.35 | **95.20** | 44.70 | 90.71 | 28.41 | 94.60 | 94.21 |
| KNN | 24.53 | 95.96 | 25.29 | 95.69 | 25.55 | 95.26 | 27.57 | 94.71 | 50.90 | 89.14 | 30.77 | 94.15 | 94.21 |
| VOS | 15.69 | 96.37 | 27.64 | 93.82 | 30.42 | 94.87 | 32.68 | 93.68 | 37.95 | **91.78** | 27.88 | 94.10 | 93.96 |
| NPOS | **5.61** | **97.64** | **4.08** | 97.52 | **14.13** | 94.92 | **8.39** | 94.67 | **18.57** | 91.35 | **10.16** | **95.22** | 93.86 |

**ImageNet-100.**    The results are shown in Table 6. All methods are trained on ResNet-101 using the ImageNet-100 dataset. We use a slightly larger model capacity to accommodate for the larger-scale dataset with high-solution images. NPOS significantly outperforms the best baseline KNN by 18.96% in FPR95. For the post-hoc OOD detection methods, we report the results by training the

Table 5: OOD detection performance on CIFAR-100 as ID. All methods are trained on ResNet-34. Values are percentages. **Bold** numbers are superior results.

| Methods | OOD Datasets | | | | | | | | | | | | ID ACC |
| | SVHN | | Places365 | | LSUN | | iSUN | | Texture | | Average | | |
| | FPR95 | AUROC | FPR95 | AUROC | FPR95 | AUROC | FPR95 | AUROC | FPR95 | AUROC | FPR95 | AUROC | |
|---|---|---|---|---|---|---|---|---|---|---|---|---|---|
| MSP | 85.30 | 72.41 | 73.40 | 81.09 | 85.55 | 74.00 | 88.55 | 68.59 | 86.45 | 71.32 | 83.85 | 73.48 | 73.12 |
| ODIN | 89.50 | 76.13 | 41.50 | 91.60 | 74.70 | 83.93 | 90.20 | 68.27 | 85.75 | 73.17 | 76.33 | 78.62 | 73.12 |
| Energy | 89.15 | 78.16 | 44.15 | 90.85 | 81.85 | 80.57 | 90.35 | 68.18 | 84.30 | 73.86 | 77.96 | 78.32 | 73.12 |
| GradNorm | 91.05 | 67.13 | 55.72 | 86.09 | 97.80 | 44.21 | 89.71 | 58.23 | 96.20 | 52.17 | 86.10 | 61.57 | 73.12 |
| ViM | 54.30 | 88.85 | 84.70 | 74.64 | 57.15 | 88.17 | 56.65 | 87.13 | 86.00 | 71.95 | 67.76 | 82.15 | 73.12 |
| KNN | 66.38 | 83.76 | 79.17 | 71.91 | 70.96 | 83.71 | 77.83 | 78.85 | 88.00 | 67.19 | 76.47 | 77.08 | 73.12 |
| VOS | 76.55 | 75.68 | **29.95** | **94.02** | 75.61 | 76.84 | 83.64 | 71.46 | 76.94 | 76.23 | 68.18 | 78.95 | 73.69 |
| NPOS | **17.98** | **96.43** | 80.41 | 73.74 | **28.90** | **92.99** | **43.50** | **89.56** | **33.07** | **92.86** | **40.77** | **89.12** | 73.78 |

Table 6: OOD detection performance on ImageNet-100 as ID. All methods are trained on ResNet-101. Values are percentages. **Bold** numbers are superior results.

| Methods | OOD Datasets | | | | | | | | | | ID ACC |
| | iNaturalist | | Places | | SUN | | Textures | | Average | | |
| | FPR95 | AUROC | FPR95 | AUROC | FPR95 | AUROC | FPR95 | AUROC | FPR95 | AUROC | |
|---|---|---|---|---|---|---|---|---|---|---|---|
| MSP | 76.30 | 82.20 | 81.90 | 77.54 | 82.70 | 78.35 | 75.30 | 80.01 | 79.05 | 79.52 | 84.16 |
| ODIN | 53.00 | 89.52 | 70.40 | 82.77 | 66.90 | 85.01 | 48.40 | 89.19 | 59.67 | 86.62 | 84.16 |
| Energy | 72.60 | 85.31 | 69.80 | 83.15 | 74.40 | 83.96 | 63.40 | 84.80 | 70.05 | 84.31 | 84.16 |
| GradNorm | 50.82 | 84.86 | 68.27 | 74.46 | 65.77 | 77.11 | 40.48 | 88.17 | 56.33 | 81.15 | 84.16 |
| ViM | 72.40 | 84.88 | 76.20 | 81.54 | 73.80 | 83.99 | 22.20 | 95.63 | 61.15 | 86.51 | 84.16 |
| KNN | 56.96 | 86.98 | 64.54 | 83.68 | 63.04 | 85.37 | 15.83 | 96.24 | 50.09 | 88.07 | 84.16 |
| VOS | 54.68 | 89.74 | 63.71 | **88.97** | **41.67** | 91.54 | 67.92 | 84.34 | 56.99 | 88.65 | 84.39 |
| NPOS | **19.49** | **96.43** | **56.17** | 88.47 | 44.91 | **91.62** | **3.97** | **99.18** | **31.13** | **93.93** | 84.23 |

model with the cross-entropy loss for 100 epochs using stochastic gradient descent with momentum 0.9. The start learning rate is 0.1 and decays by a factor of 10 at epochs 50, 75, and 90 respectively. The batch size is set to 512.

Our results above demonstrate that NPOS can achieve strong OOD detection performance without necessarily relying on the pre-trained models. Thus, our framework provides strong generality across both scenarios: training from scratch or fine-tuning on pre-trained models.

Table 7: Configurations of NPOS: training from scratch

| | CIFAR-10 | CIFAR-100 | ImageNet-100 |
|---|---|---|---|
| Training epochs | 500 | 500 | 500 |
| Momentum | 0.9 | 0.9 | 0.9 |
| Batch size | 256 | 256 | 512 |
| Weight decay | 0.0001 | 0.0001 | 0.0001 |
| Classification branch initial LR | 0.5 | 0.5 | 0.1 |
| Initial LR for $R_{\text{open}}(g)$ | 0.05 | 0.05 | 0.01 |
| LR schedule | cosine | cosine | cosine |
| Prototype update factor $\gamma$ | 0.95 | 0.95 | 0.95 |
| Regularization weight $\alpha$ | 0.1 | 0.1 | 0.1 |
| starting epoch of regularization | 200 | 200 | 200 |
| Queue size (per class) $|Q_c|$ | 600 | 600 | 1000 |
| $k$ in nearest neighbor distance | 300 | 300 | 400 |
| Number of boundary samples (per class) $m$ | 200 | 200 | 300 |
| $\sigma^2$ | 0.1 | 0.1 | 0.1 |
| Temperature $\tau$ | 0.1 | 0.1 | 0.1 |

# E    ADDITIONAL EXPERIMENTS ON MODEL CALIBRATION AND DATA-SHIFT ROBUSTNESS

**Data-shift robustness.** We evaluate the NPOS-trained model (from scratch with 5 random seeds) on different test data with distribution shifts in Table 8. Specifically, we report the mean classification accuracy and the standard deviation for measuring data-shift robustness. The number in

the bracket of the first column indicates clean accuracy on the in-distribution test data. The results demonstrate that compared to the vanilla classifier trained with the cross-entropy loss only, NPOS does not incur substantial change in distributional robustness.

Table 8: Evaluations on data-shift robustness (numbers are in %).

| ID data | Original test ACC | shifted dataset | CE-OOD ACC ($R_{closed}(g)$) | NPOS-OOD ACC ($R_{closed}(g) + R_{open}(g)$) |
|---|---|---|---|---|
| CIFAR-10 | 94.06 | CIFAR-10-C | $72.97 \pm 0.36$ | $72.63 \pm 0.52$ |
| CIFAR-100 | 74.86 | CIFAR-100-C | $46.68 \pm 0.24$ | $46.93 \pm 0.64$ |
| ImageNet-100 | 84.22 | ImageNet-C | $33.49 \pm 0.34$ | $34.29 \pm 0.76$ |
| ImageNet-100 | 84.22 | ImageNet-R | $28.98 \pm 0.62$ | $29.50 \pm 1.43$ |
| ImageNet-100 | 84.22 | ImageNet-A | $13.92 \pm 1.46$ | $12.16 \pm 2.24$ |
| ImageNet-100 | 84.22 | ImageNet-v2 | $72.17 \pm 0.95$ | $71.67 \pm 1.54$ |
| ImageNet-100 | 84.22 | ImageNet-Sketch | $18.83 \pm 0.24$ | $17.94 \pm 1.54$ |

**Model calibration.** In Table 9, we also measure the calibration error of NPOS (with 5 random seeds) on different datasets using Expected Calibration Error (ECE, in %) (Guo et al., 2017). In the implementation, we adopt the codebase[1] for metric calculation. The results suggest that NPOS maintains an overall comparable (in some cases even better) calibration performance, while achieving a much stronger performance of OOD uncertainty estimation.

Table 9: Calibration performance (numbers are in %).

| Model | Dataset | Method | ECE |
|---|---|---|---|
| training from scratch | CIFAR-10 | $CE(R_{closed}(g))$ | $0.62 \pm 0.04$ |
| | | NPOS ($R_{closed}(g) + R_{open}(g)$) | $0.27 \pm 0.06$ |
| | CIFAR-100 | CE | $3.19 \pm 0.05$ |
| | | NPOS | $4.02 \pm 0.13$ |
| | ImageNet-100 | CE | $7.17 \pm 0.07$ |
| | | NPOS | $7.52 \pm 0.29$ |
| w/ pre-trained model | ImageNet-100 | CE | $2.34 \pm 0.01$ |
| | | NPOS | $1.06 \pm 0.03$ |
| | ImageNet-1K | CE | $3.42 \pm 0.02$ |
| | | NPOS | $2.66 \pm 0.07$ |

# F  ADDITIONAL ABLATIONS ON HYPERPARAMETERS AND DESIGNS

In this section, we provide additional analysis of the hyperparameters and designs of NPOS. For all the ablations, we use the ImageNet-100 dataset as the in-distribution training data, and fine-tune on ViT-B/16.

**Ablation on the number of boundary samples.** We show in Table 10 the effect of $m$ — the number of boundary samples selected per class. We vary $m \in \{100, 150, 200, 250, 300, 350, 400\}$. We observe that NPOS is not sensitive to this hyperparameter.

Table 10: Ablation study on the number of boundary samples (per class).

| $m$ | FPR95 | AUROC | AUPR | ID ACC |
|---|---|---|---|---|
| 100 | 10.63 | 98.14 | 97.56 | 93.97 |
| 150 | 9.94 | 98.21 | 97.75 | 94.02 |
| 200 | 8.52 | 98.34 | 97.93 | 93.78 |
| 250 | 7.41 | 98.49 | 98.25 | 94.42 |
| **300** | **6.12** | **98.70** | **98.49** | 94.46 |
| 350 | 8.77 | 98.15 | 97.75 | 94.00 |
| 400 | 7.43 | 98.51 | 98.21 | 94.52 |

**Ablation on the number of samples in the class-conditional queue.** In Table 11, we investigate the effect of ID queue size $|Q_c| \in \{1000, 1500, 2000, 2500, 3000\}$. Overall, the OOD detection performance of NPOS is not sensitive to the size of the class-conditional queue. A sufficiently large $|Q_c|$ is desirable since the non-parametric density estimation can be more accurate.

[1] https://github.com/gpleiss/temperature_scaling

Table 11: Ablation study on the size of ID queue (per class).

| $|Q_c|$ | FPR95 | AUROC | AUPR | ID ACC |
|---|---|---|---|---|
| 1000 | 7.18 | 98.51 | 98.24 | 94.40 |
| **1500** | **5.76** | **98.74** | **98.73** | 94.76 |
| 2000 | 6.76 | 98.64 | 98.35 | 94.76 |
| 2500 | 8.75 | 98.28 | 97.89 | 94.36 |
| 3000 | 7.57 | 98.45 | 98.19 | 94.28 |

**Ablation on the number of candidate outliers sampled from the Gaussian kernel (per boundary ID sample).** As shown in Table 12, we analyze the effect of $p$ — the number of synthesized candidate outliers using Equation 5 around each ID boundary sample. We vary $p \in \{600, 800, 1000, 1200, 1400\}$. A reasonably large $p$ helps provide a meaningful set of candidate outliers to be selected.

Table 12: Ablation on the number of candidate outliers drawn from the Gaussian kernel.

| $p$ | FPR95 | AUROC | AUPR | ID ACC |
|---|---|---|---|---|
| 600 | 19.26 | 96.25 | 94.66 | 94.20 |
| 800 | 10.75 | 97.96 | 97.44 | 94.24 |
| **1000** | **5.76** | **98.74** | **98.73** | 94.76 |
| 1200 | 8.72 | 98.28 | 97.90 | 94.34 |
| 1400 | 10.59 | 98.12 | 97.61 | 94.38 |

**Ablation on the temperature for ID embedding optimization.** In Table 13, we ablate the effect of temperature $\tau$ used for the ID embedding optimization loss (*cf.* Equation 8).

Table 13: Ablation study on the temperature $\tau$.

| $\tau$ | FPR95 | AUROC | AUPR | ID ACC |
|---|---|---|---|---|
| 5 | 13.83 | 97.33 | 96.65 | 94.80 |
| 6 | 10.32 | 98.12 | 97.32 | 94.64 |
| 7 | 6.26 | 98.61 | 98.40 | 94.58 |
| **8** | **5.76** | 98.74 | **98.73** | 94.76 |
| 9 | 9.99 | 98.09 | 97.59 | 93.58 |
| 10 | 8.92 | **98.96** | 97.96 | 93.30 |

**Ablation on the starting epoch of adding $R_{\mathbf{open}}(g)$.** In Table 14, we ablate on the effect of the starting epoch of adding $R_{\mathrm{open}}(g)$ in training. The table shows that adding $R_{\mathrm{open}}(g)$ at the beginning of the training yields a slightly worse OOD detection performance. The reason might be that the representations are still not well-formed at the early stage of training. Instead, adding regularization in the middle of training yields more desirable performance.

Table 14: Ablation study on the starting epoch of adding $R_{\mathrm{open}}(g)$.

| epoch | FPR95 | AUROC | ID ACC |
|---|---|---|---|
| 0 | 9.36 | 98.03 | 94.06 |
| 5 | 5.79 | 98.61 | 94.21 |
| **10** | **5.76** | **98.74** | 94.76 |
| 15 | 16.21 | 97.34 | 94.39 |
| 20 | 32.68 | 94.62 | 94.16 |

**Ablation on the density estimation implementation.** NPOS adopts a class-conditional approach for outlier synthesis. For instance, it identifies the boundary ID samples by calculating the $k$-NN distance between sample pairs holding the same class label. After synthesizing the outliers in the feature space, it rejects synthesized outliers that have lower $k$-NN distance, which is also implemented in a class-conditional way. In this ablation, we contrast with an alternative class-agnostic implementation, *i.e.*, we calculate the $k$-NN distance between samples across all classes in the training set. Under the same training and inference setting, the class-agnostic NPOS gives a similar OOD detection performance compared to the class-conditional NPOS (Table 15).

Table 15: Ablation on different implementations of the non-parametric density estimation.

| Methods | OOD Datasets | | | | | | | | | | ID ACC |
| | iNaturalist | | SUN | | Places | | Textures | | Average | | |
| | FPR95 | AUROC | FPR95 | AUROC | FPR95 | AUROC | FPR95 | AUROC | FPR95 | AUROC | |
|---|---|---|---|---|---|---|---|---|---|---|---|
| Class-conditional | **0.70** | 99.14 | **9.22** | **98.48** | **5.12** | **98.86** | **8.01** | **98.47** | **5.76** | **98.74** | 94.76 |
| Class-agnostic | 2.46 | **99.32** | 11.63 | 98.06 | 8.43 | 97.69 | 9.43 | 98.45 | 7.99 | 98.38 | 94.61 |

## G    ADDITIONAL RESULTS ON THE MEAN AND STANDARD DEVIATIONS

We repeat the training of our method on ImageNet-100 with pre-trained ViT-B/16 for 5 different times. We report the mean and standard deviations for both NPOS (ours) and the most relevant baseline VOS in Table 16. NPOS is relatively stable, and outperforms VOS by a significant margin.

Table 16: Results on the mean and standard deviations after 5 runs.

| Method | FPR95 | AUROC | ID ACC |
|---|---|---|---|
| VOS | 18.26±1.1 | 95.76±0.7 | 94.51±0.3 |
| NPOS | **7.43±0.8** | **98.34±0.6** | 94.38±0.2 |

## H    SOFTWARE AND HARDWARE

We use Python 3.8.5 and PyTorch 1.11.0, and 8 NVIDIA GeForce RTX 2080Ti GPUs.

