# OpenReview forum: "Non-parametric Outlier Synthesis"
_ICLR.cc/2023/Conference — ICLR 2023 poster_

### Official Review · Reviewer_M7jy · 2022-10-22

**Confidence:** 4
**Correctness:** 3
**Technical Novelty And Significance:** 2
**Empirical Novelty And Significance:** 2
**Recommendation:** 6

**Clarity, Quality, Novelty And Reproducibility:**

The paper is clear, the framing of the story has some problems (as pointed out in the weaknesses). Although the paper is not extremely novel, the proposed approach is principled and reasonable, and clearly described, the ablations are interesting and consider relevant aspects. The information provided seems enough to reproduce the results.

**Strength And Weaknesses:**

Strengths:
- Although not extremely novel (which is not a big problem for me), the method is simple and easy to implement. The authors provide a clear algorithm box in appendix.
- The method is heuristic but principled, and illustrated with a simple and immediate diagram
- The writing is mostly clear, although the structure of the paper could be improved
- The ablations are interesting and poke interesting aspects

Weaknesses:
- Q1: Many times the performance improvements over the baselines seem marginal when measured with some metrics. Could the authors report means and standard deviations for 5-seed experiments for all the proposed methods?
- Q2: (Very important) The story of the paper is about out-of-distribution detection. However, the experiments in which the network is trained from scratch on the in-distribution data are marginal and not extensive enough (see Q3, Q4, Q5). Using a pre-trained model that has been trained on web-scale data is not particularly meaningful for the plain out-of-distribution (or open set) detection (although recent literature does it, neglecting the definition itself of the problem) since the network already knows how to extract representations about the out-of-distribution data, that make the task of distinguishing in-distribution from out-of-distribution much easier and also ill-posed. It's important for the literature to stop pretending pre-training has no impact just because this allows to report inflated numbers. For the way you frame your paper, I would not suggest putting so much emphasis on large pre-trained models and not on trained-from-scratch ones. If you want to focus on pre-trained models fine-tuning then you should probably change the framing of the paper.
- Q3: The training setup for experiments training from scratch is also pretty bizarre (why training a ResNet34 for 600 epochs? it seems like it's overtraining) Could you please justify why?
- Q4: The used baselines are not among the best performing. Could you please compare also with KFAC-LLLA [1], SNGP [2], Deep ensembles [3],, Generalized-ODIN [4] and BatchEnsemble [5]?
- Q5: To assess the goodness of your method, could you please at least show results of training from scratch on CIFAR-10 and ImageNet on a ResNet and some other architecture of your choice?
- Q6: The proposed method is applied only for part of the training. I can imagine this is to prevent the second term of the loss from compressing the in-distribution features at the early stages of training when the representations are still not well-formed. Could the authors further discuss this point and provide experiments about it? Especially because the way your method affect representations yields to Q7.
- Q7: Given the behaviour of the model, I would expect the proposed procedure might potentially have adverse effects on the calibration of a model and on its data-shift robustness . Could you perform experiments measuring these quantities on distribution-shift (for ImageNet, there's plenty of datasets: ImageNet-A/R/v2/Sketches/C etc. etc., for CIFAR-100 there's only CIFAR-100-C, but for CIFAR-10 there's also CIFAR-10.1 and CIFAR-10.2) and in-distribution data? For calibration, you may use the Mean Calibration Error. No need of re-training, just test on these datasets. I don't care if the results are bad, but if something bad happens it should be reported as a limitation.


Margin Improvements:
- Adding up arrows/down arrows to metrics of the tables
- I hope in Table 4 you reported the accuracies and not the error.

[1] https://arxiv.org/abs/2002.10118
[2] https://arxiv.org/abs/2006.10108
[3] https://arxiv.org/abs/1612.01474
[4] https://arxiv.org/abs/2002.11297
[5] https://arxiv.org/abs/2002.06715

**Summary Of The Paper:**

The paper proposes a novel methodology to perform non-parametric outlier synthesis. It suggests using KNN to identify boundary points, and then generating outlier samples by sampling from a gaussian distribution centered in such samples and discarding the synthetic samples that very likely are closer to the in-distribution data. They suggest regularising the in-distribution training performance by adding a loss term for outliers. They perform some experiments and ablations.

**Summary Of The Review:**

The paper is overall interesting, but needs to significantly improve the experimental evaluations. If the authors can significantly improve the experiments I'm happy to improve the score.

------------
Post rebuttal update: The authors have satisfactorily addressed most of my concerns. I therefore update my score to 6, inviting the authors to integrate the new experiments and comments in the new draft.

---

> ### Author Response · Authors · 2022-11-11
> **Response to Reviewer M7jy-part I**
>
> We are glad that the reviewer finds our work principled and easy to implement, with interesting ablations and a clear presentation. We thank the reviewer for the thorough comments and suggestions, which we address below:
>
> **A1. Report means and standard deviations**
>
> As suggested, we have performed training 5 different times and report the mean and standard deviations for both NPOS (ours) and the most relevant competing method VOS. NPOS outperforms the baseline by a significant margin. We attach the comparison below for your convenience. The results have also been added to our manuscript (Section G of the appendix).
>
>
> | Method | FPR95 | AUROC | ID ACC |
> | ------ | ----- | ----- | ------ |
> | VOS   | ${18.26}^{\pm 1.1}$ | ${95.76}^{\pm 0.7}$ | ${94.51}^{\pm 0.3}$  |
> | NPOS   |  **${7.43}^{\pm 0.8}$** | **${98.34}^{\pm 0.6}$** | ${94.38}^{\pm 0.2}$  |
>
>
> **A2. Using pre-trained vs. training from scratch**
> - First, **our core methodological framing and fundamental contribution (cf. Section 3) is indeed general, and has no dependency on the type of model initialization**. Our proposed method focuses on the learning algorithm and training process, rather than the starting point of the model. There is no single mention or emphasis on the pre-trained model in our method description.
>
> - Moreover, we view it as a strength that _our learning framework can be compatible with both scenarios_. As recognized by other reviewers, our extensive results provide further support for both the pre-trained model and training from scratch (cf. Section 4.2 and Section 4.4). To put this in context, we have not seen any other OOD detection work that provides such comprehensive evaluations under both scenarios as we do. With that being said, we do acknowledge some of your suggested experiments on adding CIFAR-10 and ImageNet can strengthen our paper; please see **A5** for details.
>
> - Lastly, while the debate of whether using pre-trained model is independent of our core contribution, we would like to share a few scientific reasons why it can be valuable for the research field:
>
>     - Despite the rise of pre-trained models, their safety risks of them can be inherited by all the adapted models downstream. While OOD risk has been extensively studied in the context of traditional supervised learning (i.e., optimizing a model from scratch), the regime of using a pre-trained model is much less studied. Instead of shying away from this, the research field needs to embrace this paradigm shift and develop a further understanding of its capability and incapability. Our effort contributes towards this bigger scientific quest for the research field.
>     - Using the pre-trained models allows us to better understand the OOD detection algorithms for emerging neural network architectures such as ViTs, which are typically hard to be trained from scratch. This family of model architecture is much less explored compared to the well-studied CNN model family such as ResNet. To the best of our knowledge, our work provides a comprehensive re-evaluation of recent competitive OOD detection methods (most of which were originally developed in the context of CNNs), now in the context of ViT. We believe this is valuable for the field.
>     - Despite the pre-trained model may come with good initial representation, it does not necessarily translate into perfect OOD detection performance downstream. We show that the learning algorithm plays a more fundamental role here. This is validated by our experiments in Table 3, where we contrast with [1] --- a latest development using the same pre-trained model as ours. On ViT-B/16, NPOS reduces the FPR95 from 41.87% to 6.15% (ours) — an absolute **35.72**% improvement. On ResNet, we observe an even larger improvement ($\uparrow$**51**% in FPR95). _Since both methods employ the same model initialization and test-time OOD score, the performance gap signifies the insufficiency of good initialization alone, and the importance of our proposed training loss_.
>
> [1] Fort et al. Exploring the limits of out-of-distribution detection. Advances in Neural Information Processing Systems, 34:7068–7081, 2021.
>
> **A3. Clarification on training epochs**
> When training from scratch, our loss can be viewed as prototypical representation learning, where each in-distribution sample is pushed toward the corresponding class centroid. Our training epoch follows the **common practice** in literature, which typically trains the model for 500-1000 epochs (see the configuration in [2] for example). Fine-tuning, on the other hand, allows much more computational efficiency.
>
> [2] Li, Junnan et al. "Prototypical Contrastive Learning of Unsupervised Representations." In International Conference on Learning Representations. 2021.

---

> > ### Author Response · Authors · 2022-11-11
> > **Response to Reviewer M7jy-part II**
> >
> > **A4. Justification of baselines**
> > Thank you for the suggestions. We would like to refer the reviewer to a latest work [3], which shows that KNN [4] is near SOTA among 30+ methods (including Bayesian approaches). Hence, we believe our comparisons are informative since we already included the best-performing baselines, along with Vim and VOS (the most relevant baseline to ours)---all of which are published in 2022.
> >
> >
> > [3] Yang, J. et al. OpenOOD: Benchmarking Generalized Out-of-Distribution Detection. In Thirty-sixth Conference on Neural Information Processing Systems Datasets and Benchmarks Track. 2022.
> >
> > [4] Sun, Y. et al. Out-of-distribution detection with deep nearest neighbors. In Proceedings of the International Conference on Machine Learning, pp. 20827– 20840, 2022.
> >
> > **A5. Additional experiments: training from scratch**
> > As suggested, we have conducted additional comparisons on CIFAR-10 and ImageNet-100, both trained from scratch. These new results have been incorporated in our revised draft as well; see Section D in the appendix.
> >
> > For CIFAR-10, the results are shown in the table below. All methods are trained on ResNet-18. The average FPR95 of NPOS is **9.57**%, significantly outperforming the best baseline (VOS, 27.88%).
> >
> > #### CIFAR-10
> > |        | SVHN|         | LSUN |       |iSUN |        |Texture |     | Places365|   | Average |    | ID ACC |
> > | ------ | ----- | ----- |----- | ----- |----- | ----- |----- | ----- |----- | ----- |----- |----- |----- |
> > | Method | FPR95 | AUROC |FPR95 |  AUROC |FPR95 | AUROC |FPR95 | AUROC |FPR95 | AUROC |FPR95 | AUROC |  |
> > | MSP    |  59.66 |91.25 |45.21 |93.80 |54.57 |92.12|66.45 |88.50 |62.46 |88.64|57.67| 90.86 | 94.21 |
> > | ODIN   | 20.93 |95.55 |7.26 | **98.53** |33.17| 94.65 |56.40 |86.21|63.04| 86.57|36.16| 92.30 | 94.21 |
> > | Energy  |54.41 |91.22| 10.19| 98.05|27.52 |**95.59**|55.23| 89.37|42.77 |91.02|38.02 |  93.05| 94.21 |
> > |GradNorm |  80.86|81.41 |53.87  |88.39| 60.32 |88.00 | 71.66 |80.79  | 80.71 | 72.57 |69.49 | 82.23  | 94.21 |
> > | ViM |24.95 | 95.36  | 18.80 | 96.63 | 29.25 | 95.10| 24.35 | **95.20**|44.70 | 90.71|28.41 | 94.60 | 94.21 |
> > | KNN | 24.53 |95.96 |25.29 |95.69|25.55 |95.26| 27.57 |94.71|50.90 |89.14|30.77 |94.15 |94.21 |
> > |VOS  | 15.69 | 96.37 |27.64|93.82|30.42|94.87|32.68|93.68|37.95|**91.78**|27.88|94.10|93.96|
> > | NPOS   | **4.96** | **97.15** |**3.94**|97.67|**13.69**|95.01|**7.64**|94.92|**17.61**|91.29|**9.57**|**95.20**|94.06|
> >
> > #### ImageNet
> > The results are shown in the table below for ImageNet-100. All methods are trained on ResNet-101 (we use a larger model capacity here to accommodate for this larger-scale dataset). NPOS significantly outperforms the best baseline by **14.12**% in FPR95.
> >
> >
> > |          | iNaturalist |       | Places   |       |  SUN|       | Textures |       | Average |   |ID ACC    |
> > | -------- | ----------- | ----- | ----- | ----- | -------- | ----- | ------ | ----- | ------- | ----- | -------- |
> > | Method   | FPR95       | AUROC | FPR95 | AUROC | FPR95    | AUROC | FPR95  | AUROC | FPR95   | AUROC | |
> > | MSP      |         76.30 | 82.20      |     81.90 | 77.54       |         82.70 | 78.35             |       75.30 | 80.01       |  79.05 | 79.52     | 84.16|
> > | ODIN     |             53.00 | 89.52       |       70.40 | 82.77       |        66.90 | 85.01       |           48.40 | 89.19    | 59.67 | 86.62         |   84.16 |
> > | Energy   |            72.60 | 85.31      |       69.80 | 83.15      |        74.40 | 83.96        |       63.40 | 84.80       |         70.05 | 84.31    |  84.16 |
> > | GradNorm |             50.82 | 84.86      |       68.27 | 74.46        |       65.77 | 77.11    |       40.48 | 88.17        |       56.33 | 81.15     | 84.16  |
> > | ViM      |          72.40 | 84.88      |      76.20 | 81.54       |        73.80 | 83.99       |              22.20 | 95.63     |    61.15 | 86.51      |  84.16 |
> > | KNN      |           56.96 | 86.98      |      64.54 | 83.68       |          63.04 | 85.37      |        **15.83** | **96.24**      |        50.09 | 88.07        |84.16  |
> > | VOS      | 54.68       | 89.74 | 63.71 | 88.97 | 41.67    | 91.54 | 67.92  | 84.34 | 56.99   |  88.65 |84.39 |
> > | NPOS     | **43.51**       | **91.41** | **37.84** | **92.77** | **16.28**    | **95.84** | 46.24  | 89.97 | **35.97**   | **92.50** |84.22 |

---

> > > ### Author Response · Authors · 2022-11-11
> > > **Response to Reviewer M7jy-part III**
> > >
> > > **A6. Using NPOS in the early stage of training**
> > > Great question raised! We agree that this ablation would be meaningful to include. We have conducted the suggested experiment, by introducing $R_\text{open}(g)$ earlier in the training. Please see the results below on the ImageNet-100 dataset. This has been added to our manuscript (Section F).
> > >
> > > | Starting epoch for adding $R_\text{open}(g)$ | FPR95 | AUROC | ID ACC |
> > > | ------ | ----- | ----- | ------ |
> > > | 0      | 9.36  | 98.03 | 94.06   |
> > > | 5      | **5.79**  | 98.61 | 94.21   |
> > > | 10     | 6.15  | **98.70** | 94.46   |
> > > | 15     | 16.21 | 97.34 | 94.39   |
> > > | 20     | 32.68 | 94.62 | 94.16   |
> > >
> > > The table shows that adding $R_{\text{open}}(g)$ at the beginning of the training yields a slightly worse OOD detection performance. As you concurred, this is because representations are still not well-formed at the early stage of training.
> > >
> > > **A7. Calibration and data-shift robustness**
> > > - **Data-shift robustness**. Following the suggestion, we evaluate NPOS (trained from scratch) on different test data with distribution shifts. Specifically, we report the classification accuracy for measuring data-shift robustness. The number in the bracket of the first column indicates clean accuracy on the in-distribution test data. The table below demonstrates that NPOS does not incur substantial change in distributional robustness, compared to the vanilla classifier trained with the cross-entropy loss only. The results and discussion have been updated in the manuscript (Section E).
> > >
> > >
> > > | ID dataset         |Original test ACC  | shifted dataset | CE-OOD ACC ($R_\text{closed}$) | NPOS-OOD ACC ($R_\text{closed} + R_\text{open}$) |
> > > | -------------------- | --------------- |--------------- | ------ | -------- |
> > > | CIFAR- 10 | 94.06     | CIFAR-10-C      | 73.67  | 72.96    |
> > > | CIFAR-100 |74.86    | CIFAR-100-C     | 46.69  | 46.38    |
> > > | ImageNet-100 |84.22 | ImageNet-C      | 33.64  | 35.02    |
> > > | ImageNet-100 |84.22 | ImageNet-R      | 28.94  | 32.50    |
> > > | ImageNet-100 |84.22 | ImageNet-A      | 14.69  | 9.78     |
> > > | ImageNet-100 |84.22| ImageNet-v2     | 72.61  | 69.61    |
> > > | ImageNet-100 |84.22 | ImageNet-Sketch | 18.65  | 17.42    |
> > >
> > >
> > > - **Model calibration**. We also measure the calibration error of NPOS on different in-distribution datasets using Expected Calibration Error (ECE, in %) [5]. In the implementation, we adapt the codebase [6] for metric calculation. The results suggest that NPOS maintains an overall comparable (in some cases even better) calibration performance, while achieving a much stronger performance of OOD uncertainty estimation. The results and discussion have been updated in the manuscript (Section E).
> > >
> > > | Model                 | Dataset      | Method | ECE  |
> > > | --------------------- | ------------ | ------ | ---- |
> > > | training from scratch | CIFAR-10     | CE ($R_\text{closed}$)     | 0.6  |
> > > |                       |              | NPOS ($R_\text{closed} + R_\text{open}$)   | 0.4  |
> > > |                       | CIFAR-100    | CE     | 3.1  |
> > > |                       |              | NPOS   | 4.2  |
> > > |                       | ImageNet-100 | CE     | 7.2  |
> > > |                       |              | NPOS   | 7.6  |
> > > | w pre-trained model   | ImageNet-100 | CE     | 2.3  |
> > > |                       |              | NPOS   | 1.0  |
> > > |                       | ImageNet-1k  | CE     | 3.4  |
> > > |                       |              | NPOS   | 2.6  |
> > >
> > >
> > >
> > >
> > > [5] Guo, C. et al. "On calibration of modern neural networks." International conference on machine learning. PMLR, 2017.
> > >
> > > [6] https://github.com/gpleiss/temperature_scaling

---

> ### Comment · Reviewer_M7jy · 2022-11-16
> **Thanks for the responses**
>
> I thank the authors for the responses, most of my concerns have been addressed. I'll update the score accordingly.
>
> A few minor comments (not influencing the score):
>
> - The calibration results are mixed, sometimes worse, sometimes better (which is fine). Reporting 5 seeds mean and std would be recommended.
> - The distribution shift results seem mixed, but this time sometimes the difference could be significant from the reported numbers (which is fine). Without reporting the 5 seeds mean and std it is difficult to judge.
> - The model initialization might not influence the methodology but influences the problem definition. It has already been observed that models pre-trained on ImageNet-21K and fine-tuned on ImageNet-1K can perform particularly well on ImageNet-O simply because ImageNet-O is sampled from ImageNet-21K. Therefore, it's important to consider initialization has an impact on it.

---

> > ### Author Response · Authors · 2022-11-16
> > **Thanks for your follow up**
> >
> > Dear Reviewer M7jy,
> >
> > Thank you very much for carefully reading our response. We appreciate the additional feedback, which helps us strengthen the work. We will make sure to incorporate the suggested changes (e.g., reporting results for distribution robustness & calibration with 5 seeds, and discussion on the impact of initialization) in the revised manuscript.
> >
> > Thanks again!
> >
> > Authors

---

> > ### Author Response · Authors · 2022-11-23
> > **Thanks for your follow up**
> >
> > Dear reviewer M7jy,
> >
> > To follow up, we have finished the additional experiments on distributional robustness & calibration (with 5 random seeds). The new results are attached below and have been incorporated in our draft as well (**Table 8 & Table 9**). We also revised our draft and acknowledged that model initialization has an impact on OOD detection in **Section 4.4**.
> >
> > Should there be any further concern before you update the score, we would be happy to clarify.
> >
> > - **Data-shift robustness**.
> > | ID dataset   | Original test ACC | shifted datasets | CE-OOD ACC ($R_\text{closed}$) | NPOS-OOD ACC ($R_\text{closed} + R_\text{open}$) |
> > | ------------ | ----------------- | ---------------- | ------------------------------ | ------------------------------------------------ |
> > | CIFAR- 10    | 94.06             | CIFAR-10-C       | 72.97 $\pm$ 0.36               | 72.63 $\pm$ 0.52                                 |
> > | CIFAR-100    | 74.86             | CIFAR-100-C      | 46.68 $\pm$ 0.24               | 46.93 $\pm$ 0.64                                 |
> > | ImageNet-100 | 84.22             | ImageNet-C       | 33.49 $\pm$ 0.34               | 34.29 $\pm$ 0.76                                 |
> > | ImageNet-100 | 84.22             | ImageNet-R       | 28.98 $\pm$ 0.62               | 29.50 $\pm$ 1.43                                 |
> > | ImageNet-100 | 84.22             | ImageNet-A       | 13.92 $\pm$ 1.46               | 12.16 $\pm$ 2.24                                 |
> > | ImageNet-100 | 84.22             | ImageNet-v2      | 72.17 $\pm$ 0.95               | 71.67 $\pm$ 1.54                                 |
> > | ImageNet-100 | 84.22             | ImageNet-Sketch  | 18.83 $\pm$ 0.24               | 17.94 $\pm$ 1.54                                 |
> > - **Model Calibration**.
> > | Model                 | Dataset      | Method                                   | ECE            |
> > | --------------------- | ------------ | ---------------------------------------- | -------------- |
> > | training from scratch | CIFAR-10     | CE ($R_\text{closed}$)                   | 0.62$\pm$ 0.04 |
> > |                       |              | NPOS ($R_\text{closed} + R_\text{open}$) | 0.27$\pm$ 0.06 |
> > |                       | CIFAR-100    | CE                                       | 3.19$\pm$ 0.05 |
> > |                       |              | NPOS                                     | 4.02$\pm$ 0.13 |
> > |                       | ImageNet-100 | CE                                       | 7.17$\pm$ 0.07 |
> > |                       |              | NPOS                                     | 7.52$\pm$ 0.29 |
> > | w pre-trained model   | ImageNet-100 | CE                                       | 2.34$\pm$ 0.01 |
> > |                       |              | NPOS                                     | 1.06$\pm$ 0.03 |
> > |                       | ImageNet-1k  | CE                                       | 3.42$\pm$ 0.02 |
> > |                       |              | NPOS                                     | 2.66$\pm$ 0.07 |

---

### Official Review · Reviewer_8Tpk · 2022-10-24

**Confidence:** 3
**Correctness:** 4
**Technical Novelty And Significance:** 2
**Empirical Novelty And Significance:** 3
**Recommendation:** 6

**Clarity, Quality, Novelty And Reproducibility:**

It is a natural extension to existing work on sampling outliers in feature-space. The experiments are well-done and show improvement with respect to existing approaches.

**Strength And Weaknesses:**

- The paper addresses a problem that is important and impactful.

- I am not convinced that the move from Gaussian priors on the feature space to non-parametric forms is an improvement. Existing techniques for generative models train the encoder such that the embeddings have a Gaussian distribution. A valid objection is that this Gaussian distribution can only be expected to hold over the training data. However, nonparametric models require selection of kernel functions, which generally require domain-specific knowledge of the phenomenon in question. If domain-specific knowledge can inform the kernel of the OOD distribution, why can it not inform the selection of the ID in the first place?

- A natural drawback of detecting outliers at the feature level is that a given input-to-feature mapping may do an inadequate job of describing an input that is OOD. In other words, if the OOD sample is also OOD for the the feature space, I am not sure that this approach could help much.

**Summary Of The Paper:**

When deploying ML models in the field, it is important to be able to detect when incoming data is an outlier. This can be quite difficult. This paper proposes an approach which learns to generate out-of-distribution data, and uses it to train a classifier. This classifier can then be used in deployment with the original ML model, to reject data that is too different from the training distribution of the trained model.

Previous work proposes synthesizing virtual outliers from the low-likelihood regions of the feature space, which is more tractable than synthesizing outliers in the input space, but it modeled the distribution of the feature space as class-conditional Gaussians. The authors instead propose to perform non-parametric outlier synthesis.

**Summary Of The Review:**

The work is a small increment to existing work.

---

> ### Author Response · Authors · 2022-11-11
> **Response to Reviewer 8Tpk**
>
> We are glad that the reviewer found our paper addresses an important and impactful problem with well-done experimental results. We also thank you for the constructive comments and suggestions, which we address below:
>
> **A1. Why move from Gaussian (parametric) to non-parametric?**
> Excellent question here. Firstly, we would like to clarify that NPOS operates on a _discriminative model_, which fundamentally differs from a generative model that enforces a Gaussian prior in the latent feature space. Due to the lack of explicit constraint, embeddings of a classification model _do not necessarily_ follow a Gaussian distribution. This has been previously verified in [1] using the Henze-Zirkler multivariate normality test [2].
>
> In contrast, our non-parametric approach can alleviate this assumption and thus synthesize outliers for _any embedding geometry_ including non-Gaussians. This flexibility and generality are in practice desirable.
>
> Moreover, it is well-known that the choice of kernel function form (e.g., Gaussian vs. Epanechnikov) is not crucial, while the kernel bandwidth parameter is. See for example Larry Wasserman's notes [3] Section 4. Please note that our bandwidth parameter is also analyzed in Section 4.3 (Ablation on the variance $\sigma^2$ in sampling). We observe that the performance of NPOS is _insensitive_ under a moderate variance.
>
>
> [1] Sun, Y. et al. Out-of-distribution detection with deep nearest neighbors. In Proceedings of the International Conference on Machine Learning, pp. 20827– 20840, 2022.
>
> [2] Henze, N. et al. A class of invariant consistent tests for multivariate normality. Communications in statistics theory and Methods, 19(10):3595–3617, 1990.
>
> [3] https://www.stat.cmu.edu/~larry/=sml/densityestimation.pdf
>
>
> **A2. Discussion on the role of good input-to-feature mapping**
> You raise another great point. Performing OOD detection in the feature space offers the benefit of being more tractable than in the input space, due to significantly reduced dimensionality.  As with many other feature-based OOD detection approaches, the efficacy depends on the goodness of the feature encoder.
>
> We have verified the role of ID feature mapping in **Section 4.2** (paragraph "Learning compact ID representation benefits NPOS"). Suboptimal function mapping (e.g. trained from a cross-entropy loss) can lead to worsened performance. Interestingly, we show that our loss function in Equation 8 facilitates learning a compact representation for ID data, and benefits NPOS.

---

> ### Public Comment · ~Jingyang_Zhang2 · 2023-02-07
> **I second the argument that feature extraction is important**
>
> > A natural drawback of detecting outliers at the feature level is that a given input-to-feature mapping may do an inadequate job of describing an input that is OOD. In other words, if the OOD sample is also OOD for the the feature space, I am not sure that this approach could help much.
>
> I hold the exact same point of view. By generating outliers in the feature space, yes, one can better control the final outputs for ID and OOD samples. Yet for this to really work you need to make sure that the backbone/feature extractor indeed maps the OOD inputs to the region of synthesized outliers. Imagine that a "bad" feature extractor which maps ID and OOD data to very similar representations: In this case there is no way you could separate them. I think this is why in this work the authors choose to use CLIP pre-trained models, which are already super powerful feature extractors even used as zero-shot.
>
> To me I would view this work as answering "Given a good feature extractor, how can we better separate ID and OOD in the feature space". It is a meaningful question, yet it would be even better if the authors can show that the method is also helpful when the feature extractor is not that good (e.g., without the powerful CLIP pre-training). I noticed in Sec. 4.4 there are some experiments for training from scratch, yet the results are not convincing since most of the selected OOD datasets are considerably different from the ID images (honestly I don't see why LSUN-resize and LSUN-crop are still being used for evaluation after the CSI paper (NeurIPS'20) identified that they contain obvious artifacts and simply low-level statistics can already achieve near-perfect detection), plus the method is only compared with VOS in that section.

---

### Official Review · Reviewer_fjV3 · 2022-10-25

**Confidence:** 3
**Correctness:** 4
**Technical Novelty And Significance:** 3
**Empirical Novelty And Significance:** 4
**Recommendation:** 6

**Clarity, Quality, Novelty And Reproducibility:**

This paper is novel (to the best of my knowledge) and relatively easy to follow.

**Strength And Weaknesses:**

This paper has an interesting and intuitive algorithm, and carefully validates this algorithm over a wide variety of datasets. Although their algorithm lacks theoretical guarantees, I believe it is still a very interesting and important contribution -- subsequent work could very well lead to more theoretical insights about this problem.

In addition to addressing a relevant problem, I think this paper proposes an interesting high level approach to applying non-parametric algorithms for OOD generalization, namely outlier synthesis. A key property of classical non-parametric theory is that it generalizes well within the support of a probability distribution (in the large sample limit). To have any hope of giving results on OOD samples, non-parametric methods need training data that is also ``OOD", and this paper gives a natural way of generating such data.



**Summary Of The Paper:**

This paper provides a non-parametric method for OOD generalization. They consider a particular form of OOD, where the objective is to abstain from classification on points that are considered OOD. They define the ground truth of such points according to a level set, $\mathcal{L}$, which comprises of all points with probability density (in the training distribution) at most $\beta$.

They propose a loss function that is a linear combination of the classification risk over in-distribution samples with the error rate of the OOD detector that is implicit within the classifier. At a high level, their training algorithm works as follows:

1. Use nearest neighbors to estimate the probability density of points in the training sample.
2. Use the parameter $\beta$ to create estimates on which points are outliers.
3.  Use Gaussian noise centered at randomly chosen outliers to create synthetic outliers.
4. apply rejection sampling to make the synthetic outliers conform to the desired distribution.
5. Finally, apply nearest neighbors for both classification and outlier detection.

Here, nearest neighbors is always applied in the feature space which is learned. They then validate their algorithm over extensive experiments on multiple datasets.

**Summary Of The Review:**

 I think that this work should be accepted. I am quite interested in theoretical problems that this paper naturally introduces, and I find their algorithm quite aesthetically pleasing (beyond being effective).

---

> ### Author Response · Authors · 2022-11-11
> **Response to Reviewer fjV3**
>
> We are really encouraged that you recognize our method to be novel, interesting, aesthetically pleasing, and with effective and extensive empirical results.
>
> Your summary and comments are insightful and spot-on :)
>
> **Theoretical insights**
> On the theoretical side, one advantage of our framework is the principled interpretation from a rejection sampling perspective. Rejection sampling is a long-established Monte Carlo method (see for example John von Neumann used in the 1950s [1]). In a nutshell, this framework allows sampling data from a sophisticated (or "difficult to sample from") target distribution with the help of a proxy/proposal distribution. Our method draws the connection to this classical framework, where the distribution of OOD data can be viewed as "difficult to sample from" or "difficult to characterize in the explicit distributional form". To the best of our knowledge, our work is the first to establish the connection between rejection sampling and modern OOD detection problem.
>
> Given the connection, our algorithm is thus theoretically grounded. As you said,
>
> > "I believe it is still a very interesting and important contribution -- subsequent work could very well lead to more theoretical insights about this problem."
>
> ---which we very much agree and expect more future works to delve deeper into this.
>
> Aside from the soundness of the mathematical framework, our work also contributes a non-trivial realization of the rejection sampling in modern neural networks (cf. Section 3.1). One known drawback of rejection sampling is computational inefficiency, especially in higher dimensions. Hence, we proposed an algorithm to provide an efficient surrogate by (1) identifying boundary ID samples and then (2) sampling based on the boundary samples---both components are also new for the field.
>
>
> [1] J. von Neumann, "Various techniques used in connection with random digits. Monte Carlo methods", Nat. Bureau Standards, 12 (1951), pp. 36–38.

---

### Author Response · Authors · 2022-11-11
**General Response**

We thank all the reviewers for their time and valuable comments. We are encouraged to see that **ALL** reviewers find our paper **interesting** and studies an **important** problem (fjV3, 8Tpk, M7jy), that our method is **novel**, **intuitive**, **principled** and **aesthetically pleasing** (fjV3, M7jy), and the results are **effective**, **extensive**, **carefully validated**, **well-done** with **interesting** ablations (fjV3, 8Tpk, M7jy). We are glad that reviewers found the paper **clear** and **easy to follow** (fjV3, M7jy).

We respond to each reviewer's comments in detail below. We have also revised the manuscript according to the reviewers' suggestions, and we believe this makes our paper much stronger.  The main changes are highlighted in green, which include:

+ Added results on training from scratch for CIFAR-10 and ImageNet (Section D)
+ Added mean and standard deviations (Section G).
+ Added ablations on adding NPOS loss in an earlier stage of training (Section F).
+ Added additional results on calibration (Section E)
+ Added additional evaluations on distribution-shift robustness (Section E)
+ Other changes: adding arrows to tables, changing from ID error to ID ACC.

---

### Decision · Program_Chairs · 2023-01-20

**Decision:**

Accept: poster

**Justification For Why Not Higher Score:**

The algorithm largely addresses the less challenging setting of OOD detection, in which a good embedder (e.g., from CLIP) is already provided.

**Justification For Why Not Lower Score:**

The algorithm proposed is very clean and simple to implement yet yields non-trivial benefits (most notably FPR95).

**Metareview: Summary, Strengths And Weaknesses:**

The paper considers a simple heuristic for training outlier detection systems: basically, one generates synthetic outliers by adding Gaussian perturbations with rejection sampling in a well-trained latent space (e.g., a CLIP-based embedder) and trains a classifier (by augmenting the loss appropriately) to detect the outliers. The paper was discussed in a Zoom meeting with the reviewers, and the feeling among the reviewers and the AC was fairly positive: the method is fairly clean and simple but yields non-trivial benefits for several metrics (most notably, FPR95, when compared to some simpler baselines). The simplicity would also possibly make the method amenable to theoretical analysis for future works. The discussion also concluded that the authors should be a bit more explicit that the setting this algorithm is interesting for is when a very good embedder is available: more precisely, in OOD settings in general, one can consider the situation in which the features are possibly wrong; and the setting in which the features are mostly good, but they are "weighted wrong". The algorithm in the paper addresses the latter (less challenging) setting, in which to detect the outliers, the feature space is mostly right.

**Note From Pc:**

if the above contains the word "oral" or "spotlight" please see: "oral" presentation means -> notable-top-5% and "spotlight" means -> notable-top-25%. As stated in our emails, we are disassociating presentation type from AC recommendations

**Summary Of Ac-Reviewer Meeting:**

The feeling among the reviewers and the AC was fairly positive: the method is fairly clean and simple but yields non-trivial benefits for several metrics (most notably, FPR95, when compared to some simpler baselines). The simplicity would also possibly make the method amenable to theoretical analysis for future works. The discussion also concluded that the authors should be a bit more explicit that the setting this algorithm is interesting for is when a very good embedder is available: more precisely, in OOD settings in general, one can consider the situation in which the features are possibly wrong; and the setting in which the features are mostly good, but they are "weighted wrong". The algorithm in the paper addresses the latter (less challenging) setting, in which to detect the outliers, the feature space is mostly right.